# A Kernel-based Test of Independence for Cluster-correlated Data

**Hongjiao Liu**
Department of Biostatistics
University of Washington
liuhj@uw.edu

**Anna M. Plantinga**
Department of Mathematics and Statistics
Williams College
amp9@williams.edu

**Yunhua Xiang**
Department of Biostatistics
University of Washington
xiangyh@uw.edu

**Michael C. Wu**
Public Health Sciences Division
Fred Hutchinson Cancer Research Center
mcwu@fredhutch.org

## Abstract

The Hilbert-Schmidt Independence Criterion (HSIC) is a powerful kernel-based statistic for assessing the generalized dependence between two multivariate variables. However, independence testing based on the HSIC is not directly possible for cluster-correlated data. Such a correlation pattern among the observations arises in many practical situations, e.g., family-based and longitudinal data, and requires proper accommodation. Therefore, we propose a novel HSIC-based independence test to evaluate the dependence between two multivariate variables based on cluster-correlated data. Using the previously proposed empirical HSIC as our test statistic, we derive its asymptotic distribution under the null hypothesis of independence between the two variables but in the presence of sample correlation. Based on both simulation studies and real data analysis, we show that, with clustered data, our approach effectively controls type I error and has a higher statistical power than competing methods.

## 1 Introduction

We are often interested in studying the dependence between two multivariate variables. For example, in genetic studies, we may want to assess the association between multiple genetic variants within a gene and a group of traits that likely share a common genetic mechanism [1, 2]. In microbiome studies, we may wish to investigate the association between the overall composition of human microbiota, including hundreds of microbial taxa, and multiple host metabolites from a particular metabolic pathway [3, 4]. Such multivariate analyses aggregate information across variables and are often more powerful than univariate analyses. Meanwhile, correlated observations arise in many practical situations. Family-based designs are common in genetic studies [5], where multiple family members are recruited together into the study. Longitudinal data are common in epidemiological studies [6], where variables of interest are measured on the subjects repeatedly over time. Such study designs introduce clustered dependence among the observations and require proper accommodation. In this work, we aim to develop an approach for assessing the dependence between two multivariate variables, based on cluster-correlated data.

A variety of parametric and semi-parametric methods has been proposed to study the association between one or multiple exposure variables and a multivariate longitudinal (or other cluster-correlated) outcome. These methods often extend upon existing tools for univariate longitudinal data. For example, many studies stack the multivariate outcome into a single response vector and then apply the

35th Conference on Neural Information Processing Systems (NeurIPS 2021).

usual methods that account for clustered data, such as generalized estimating equations (GEE) [7, 8] and random effects models [9, 10]. Such approaches generally apply to a low-dimensional setting [11] and are subject to limitations typical of parametric and semi-parametric methods. Random-effect-based methods require assumptions on the distribution of the multivariate outcome. GEE-based methods rely on good estimation of the correlation structure within clusters as well as across different outcome variables to achieve a high efficiency. Finally, both approaches assume parametric (often linear) relationships between the exposure and the outcome. Therefore, they can only evaluate a limited number of dependence patterns, and are not sufficient as independence tests.

Here we base our approach on the Hilbert-Schmidt Independence Criterion (HSIC), a non-parametric kernel-based measure for assessing the generalized dependence between two multivariate, and potentially high-dimensional, variables [12]. By mapping the two variables into reproducing kernel Hilbert spaces (RKHS's), the population HSIC can be viewed as a measure of maximized covariance between functions in the two RKHS's. When the RKHS's being used are characteristic [13], the population HSIC is zero if and only if the two variables are independent. This measure makes no assumption on the distributions of the variables or the nature of the dependence.

The original HSIC-based independence test [14] applies to independent and identically distributed (i.i.d.) observations. Several extensions have been made to accommodate non-i.i.d. data, but none of the tests, to our knowledge, directly applies to clustered data at an observation level. Zhang et al. (2008) [15] extended the HSIC to certain sequence data, such as XOR sequence and Gaussian process, by specifying the correlation structure of the data as a graphical model and deriving the test statistic based on the maximal cliques. Chwialkowski et al. [16] and Wang et al. [17] developed HSIC-based tests to evaluate the dependence between two time series or random processes in general. Flaxman et al. [18] considered spatial and temporal data; they proposed to first use Gaussian process regression to remove dependence on space and time from the raw variables, and then perform the HSIC test on the resulting de-correlated residuals. However, their approach generally applies to independence testing between two univariate variables. A study with an aim closest to ours is by Rudra et al. [19]: They analyzed the association between multiple genetic variants and a multivariate longitudinal outcome. Rudra et al. concatenated the outcome measurements from different time points at the subject level, and then applied the HSIC test to the subject-level data. Although this is a straightforward approach to deal with clustered correlation, there could be a loss of statistical power by analyzing data at the subject/cluster level rather than observation level.

In this work, we present the first HSIC-based independence test for cluster-correlated data. Using the empirical HSIC [14] as our test statistic, we derive its asymptotic distribution under the null hypothesis of independence between the two variables but in the presence of clustered correlation among observations. We also examine the behavior of the test statistic under the alternative hypothesis and establish the consistency of our test. Furthermore, we provide a way to approximate the asymptotic null distribution of the test statistic and allow for statistical testing in practice. In simulation studies, our proposed test controls type I error rates well and has a much higher statistical power than competing methods across a range of scenarios. In an application to a longitudinal microbiome-metabolite data set, compared to other approaches, our proposed test identifies a larger number of metabolic pathways significantly associated with the overall microbiome composition, highlighting the value of our test in scientific studies.

The remaining sections are organized as following. In Section 2, we provide our background assumption on clustered data and an overview of the HSIC statistic. In Section 3, we study the asymptotic behavior of the HSIC statistic under null and alternative hypotheses, and construct a statistical test of independence for cluster-correlated data. In Section 4, we demonstrate the performance of our proposed test on both simulated and real data. In Section 5, we summarize our work, discuss the limitations of our proposed test and provide a conclusion.

## 2   Background

In this section, we introduce our assumption on cluster-correlated data and give an overview of the HSIC statistic.

## 2.1 General setting

Let $P_{XY}$ be a probability measure defined on a sample space $\mathcal{X} \times \mathcal{Y}$, where both $\mathcal{X}$ and $\mathcal{Y}$ can be multi-dimensional. Let $P_X$ and $P_Y$ be the marginal distributions on $\mathcal{X}$ and $\mathcal{Y}$, respectively. The variables $X$ and $Y$ are statistically independent if $P_{XY} = P_X P_Y$ (equivalently, we can write $X \perp\!\!\!\perp Y$).

We consider a sample of clustered data $\{(X_j, Y_j)\}_{j=1}^n$ drawn from $P_{XY}$, where the pattern of clustered correlation is balanced and complete:

**Assumption 2.1.** *The observations* $(X_1, Y_1), \cdots, (X_n, Y_n)$ *are identically distributed according to* $P_{XY}$, *and can be divided into* $m$ *clusters of fixed size* $d$ *(i.e.,* $n = md$*). In particular, the* $m$ *clusters*

$$\left\{ \Big[ (X_{di-d+1}, Y_{di-d+1}), \cdots, (X_{di}, Y_{di}) \Big] \right\}_{i=1}^m$$

*are independent from one another while having identical within-cluster correlation structure.*

The specific correlation structure among the observations in each cluster can be arbitrary. We are interested in studying the dependence between $X$ and $Y$ based on the sample $\{(X_j, Y_j)\}_{j=1}^n$.

## 2.2 Hilbert-Schmidt Independence Criterion

We briefly review the Hilbert-Schmidt Independence Criterion (HSIC) proposed by Gretton et al. (2005a) [12]. The HSIC measures the generalized dependence between two variables $X$ and $Y$, by embedding $X$ and $Y$ into reproducing kernel Hilbert spaces (RKHS's) and maximizing the covariance between functions of $X$ and $Y$ in the RKHS's.

Let $\mathcal{H}_X$ be an RKHS on $\mathcal{X}$ with associated kernel function (i.e., inner product in the RKHS) $k_X : \mathcal{X} \times \mathcal{X} \to \mathbb{R}$, and let $\mathcal{H}_Y$ be an RKHS on $\mathcal{Y}$ with associated kernel function $k_Y : \mathcal{Y} \times \mathcal{Y} \to \mathbb{R}$. Following Gretton et al. (2007) [14], the cross-covariance operator $C_{XY} : \mathcal{H}_Y \to \mathcal{H}_X$ can be defined such that, for any $f \in \mathcal{H}_X$ and $g \in \mathcal{H}_Y$,

$$\langle f, C_{XY} g \rangle_{\mathcal{H}_X} = \mathbb{E}_{XY} \Big[ \big( f(X) - \mathbb{E}_X[f(X)] \big) \big( g(Y) - \mathbb{E}_Y[g(Y)] \big) \Big] = \mathbb{C}\text{ov} \Big( f(X), g(Y) \Big).$$

As shown by Gretton et al. (2005b) [20], the operator norm (i.e., the largest singular value) of $C_{XY}$, defined by $\|C_{XY}\| := \sup_{f \in \mathcal{H}_X, g \in \mathcal{H}_Y, \|f\|_\infty \leq 1, \|g\|_\infty \leq 1} \mathbb{C}\text{ov} \big( f(X), g(Y) \big)$, is zero if and only if $X \perp\!\!\!\perp Y$, given that the kernels $k_X$ and $k_Y$ are *universal* (see Definition 5 of [20]). In this sense, $\|C_{XY}\|$ is a measure of independence between $X$ and $Y$.

The largest singular value of $C_{XY}$ becomes zero when the sum of all squared singular values, denoted as the squared Hilbert-Schmidt norm [12], is zero. Therefore, the squared Hilbert-Schmidt norm of $C_{XY}$, $\|C_{XY}\|_{HS}^2$, is also an independence criterion. This measure is defined as the population HSIC, which can be expressed conveniently in terms of kernel functions:

$$\text{HSIC}(P_{XY}) := \|C_{XY}\|_{HS}^2 = \mathbb{E}_{XX'YY'}[k_X(X, X') k_Y(Y, Y')]$$
$$+ \mathbb{E}_{XX'}[k_X(X, X')] \mathbb{E}_{YY'}[k_Y(Y, Y')] - 2 \mathbb{E}_{XY} \Big[ \mathbb{E}_{X'}[k_X(X, X')] \mathbb{E}_{Y'}[k_Y(Y, Y')] \Big],$$

where $X'$ is an independent copy of $X$. It is obvious that, if $X$ is independent from $Y$, then we have $\text{HSIC}(P_{XY}) = 0$. Furthermore, for certain *characteristic* $k_X$ and $k_Y$ [13], $\text{HSIC}(P_{XY}) = 0$ if and only if $X \perp\!\!\!\perp Y$. Example characteristic kernels include Gaussian kernels and Laplacian kernels [21].

To estimate the population HSIC from a sample $\{(X_j, Y_j)\}_{j=1}^n$, the empirical HSIC can be used:

$$\text{HSIC}(P_n) := \frac{1}{n^2} \sum_{i,j}^n k_X(X_i, X_j) k_Y(Y_i, Y_j) + \frac{1}{n^4} \sum_{i,j,q,r}^n k_X(X_i, X_j) k_Y(Y_q, Y_r)$$
$$- \frac{2}{n^3} \sum_{i,j,q}^n k_X(X_i, X_j) k_Y(Y_i, Y_q).$$

Define the kernel matrices $\boldsymbol{K}_X$ and $\boldsymbol{K}_Y$ such that the $(i, j)$-th element of $\boldsymbol{K}_X$ is $k_X(X_i, X_j)$ and the $(i, j)$-th element of $\boldsymbol{K}_Y$ is $k_Y(Y_i, Y_j)$. Then the empirical HSIC can also be written in terms of $\boldsymbol{K}_X$ and $\boldsymbol{K}_Y$:

$$\text{HSIC}(P_n) = \frac{1}{n^2} \text{tr}(\boldsymbol{H} \boldsymbol{K}_X \boldsymbol{H} \boldsymbol{K}_Y),$$

where $\boldsymbol{H} = \boldsymbol{I} - \frac{1}{n}\boldsymbol{1}\boldsymbol{1}^T$ is a centering matrix.

Both Gretton et al. (2007) [14] and Zhang et al. (2012) [22] have derived the asymptotic distribution of $\mathrm{HSIC}(P_n)$ under the null hypothesis of independence between $X$ and $Y$ as a weighted sum of chi-square variables, when the observations are i.i.d. In Section 3, we examine the asymptotic behavior of $\mathrm{HSIC}(P_n)$ based on cluster-correlated observations. It turns out that the null distribution in this case is still a weighted sum of chi-square variables, where the weights are now modified.

## 3 HSIC for cluster-correlated data

Based on the clustered data setting in Section 2.1, we aim to test the null hypothesis $H_0 : X \perp\!\!\!\perp Y$ using the empirical HSIC statistic. We first define some useful parameters and statistics.

Assume that the kernel matrices $\boldsymbol{K}_X$ and $\boldsymbol{K}_Y$ defined in Section 2.2 are positive semi-definite. We focus our attention on the centered kernel matrices: $\widetilde{\boldsymbol{K}}_X := \boldsymbol{H}\boldsymbol{K}_X\boldsymbol{H}$ and $\widetilde{\boldsymbol{K}}_Y := \boldsymbol{H}\boldsymbol{K}_Y\boldsymbol{H}$. Let $\tilde{k}_X$ and $\tilde{k}_Y$ be the centered kernel functions [1] derived from $k_X$ and $k_Y$, with associated RKHS's $\widetilde{\mathcal{H}}_X$ and $\widetilde{\mathcal{H}}_Y$, respectively. Note that the empirical HSIC can be written as $\mathrm{HSIC}(P_n) = \frac{1}{n^2}\mathrm{tr}(\widetilde{\boldsymbol{K}}_X\widetilde{\boldsymbol{K}}_Y)$.

Let $\gamma_{X,r}$ be the $r$-th largest eigenvalue and $\boldsymbol{u}_{X,r} = (u_{X,r}(X_1), \cdots, u_{X,r}(X_n))^T$ be the $r$-th eigenvector of $\widetilde{\boldsymbol{K}}_X$. Similarly, we define the eigenvalues $\gamma_{Y,r}$'s and eigenvectors $\boldsymbol{u}_{Y,r}$'s for $\widetilde{\boldsymbol{K}}_Y$. On the other hand, let $\lambda_{X,r}$ be the $r$-th largest eigenvalue of the kernel $\tilde{k}_X$ with respect to $P_X$, with associated eigenfunction $\phi_{X,r}(\cdot)$, such that $\int \tilde{k}_X(x, x')\phi_{X,r}(x')dP_X(x') = \lambda_{X,r}\phi_{X,r}(x)$. Similarly, we define the eigenvalues $\lambda_{Y,r}$'s and eigenfunctions $\phi_{Y,r}$'s for the kernel $\tilde{k}_Y$ with respect to $P_Y$.

For any fixed $R \in \mathbb{N}$, let $\tilde{k}_{X,R}(x, x') := \sum_{r=1}^{R} \lambda_{X,r}\phi_{X,r}(x)\phi_{X,r}(x')$. For each $r$ where $\gamma_{X,r} > 0$, let $g_{X,r}(x) := \frac{\sqrt{n}}{\gamma_{X,r}}\sum_{j=1}^{n} \tilde{k}_X(x, X_j)u_{X,r}(X_j)$. Define $\tilde{k}_{Y,R}$ and $g_{Y,r}$ similarly. The upcoming theorems will rely on the following assumption:

**Assumption 3.1.** *Suppose that $\mathbb{E}[\tilde{k}_X^2(X, X')] < \infty$ and $\mathbb{E}[\tilde{k}_Y^2(Y, Y')] < \infty$. Assume that, for each $R \in \mathbb{N}$, the classes $\mathcal{C}_X := \{x \mapsto (\tilde{k}_X - \tilde{k}_{X,R})^2(x, x') : x' \in \mathcal{X}\}$ and $\mathcal{C}_Y := \{y \mapsto (\tilde{k}_Y - \tilde{k}_{Y,R})^2(y, y') : y' \in \mathcal{Y}\}$ are $P_X$-Donsker and $P_Y$-Donsker [23], respectively. Further assume that, for each $r$, the functions $x \mapsto g_{X,r}(x)$ and $y \mapsto g_{Y,r}(y)$ converge uniformly in probability as $m \to \infty$, with their limit functions in $L_2(P_X)$ and $L_2(P_Y)$, respectively.*

In general, Assumption 3.1 ensures that the data-dependent eigenvalues and (elements of) eigenvectors of the kernel matrices $\widetilde{\boldsymbol{K}}_X$ and $\widetilde{\boldsymbol{K}}_Y$ converge in probability to eigenvalues and eigenfunctions of the kernels $\tilde{k}_X$ and $\tilde{k}_Y$. We show in Appendix A.1 that the Donsker class condition in Assumption 3.1 holds for Gaussian kernels. Now we can establish the asymptotic distribution of $\mathrm{HSIC}(P_n)$ under $H_0 : X \perp\!\!\!\perp Y$ based on clustered data.

**Theorem 3.2.** *Suppose that, for two multivariate random variables $X$ and $Y$, we have centered kernels $\tilde{k}_X$ and $\tilde{k}_Y$ with discrete eigenvalues. Suppose that Assumption 2.1 and Assumption 3.1 hold. Under the null hypothesis $H_0 : X \perp\!\!\!\perp Y$, we have*

$$n\,\mathrm{HSIC}(P_n) = \frac{1}{n}\mathrm{tr}(\widetilde{\boldsymbol{K}}_X\widetilde{\boldsymbol{K}}_Y) \xrightarrow{d} \sum_{t=1}^{\infty} \ell_t z_t^2 \text{ as } m \to \infty, \tag{1}$$

*where $z_t$'s are i.i.d. standard normal variables, and $\ell_t$'s are the solutions to the eigenvalue problem*

$$\begin{aligned}&\ell_t \psi_{t,rs} \\ &= \frac{1}{d}\sum_{p,q=1}^{\infty} \mathbb{E}\left[\left(\sum_{i=1}^{d}\sqrt{\lambda_{X,r}\lambda_{Y,s}}\phi_{X,r}(X_i)\phi_{Y,s}(Y_i)\right)\left(\sum_{i=1}^{d}\sqrt{\lambda_{X,p}\lambda_{Y,q}}\phi_{X,p}(X_i)\phi_{Y,q}(Y_i)\right)\right]\psi_{t,pq}\end{aligned}$$

*for some double sequence $\{\psi_{t,rs}\}_{r,s=1}^{\infty} \in \mathbb{R}$.*

---

[1]For the kernel function $k_X$, the corresponding centered kernel function $\tilde{k}_X$ is: $\tilde{k}_X(x, x') = k_X(x, x') - \mathbb{E}_X[k_X(X, x')] - \mathbb{E}_{X'}[k_X(x, X')] + \mathbb{E}_{XX'}[k_X(X, X')]$.

The proof of Theorem 3.2 is provided in Appendix B. To prove the theorem, we first show the convergence of eigenvalues and eigenvectors of $\widetilde{\boldsymbol{K}}_X$ and $\widetilde{\boldsymbol{K}}_Y$ in the presence of clustered data. We then adopt a strategy similar to that of Zhang et al. (2012) [22]: The test statistic $n\,\mathrm{HSIC}(P_n)$ can be expressed as a sum of squared terms, $\sum_{r,s=1}^{n} Q_{rs}^2$, where the terms $Q_{rs}$'s depend on eigenvalues and eigenvectors of the kernel matrices. We could show that $Q_{rs}$'s are asymptotically jointly normal with mean zero under $H_0$, and the asymptotic variances and covariances of these terms depend on eigenvalues and eigenfunctions of the kernels $\tilde{k}_X$ and $\tilde{k}_Y$.

As a result, the asymptotic distribution of $\mathrm{HSIC}(P_n)$ under $H_0$ is a weighted sum of chi-square variables. In particular, we require the number of clusters, $m$, to be sufficiently large. Knowing the null distribution of the test statistic enables us to construct a statistical test at a given significance level. To examine the power of the proposed test, we further explore the behavior of the test statistic when the null hypothesis is violated. The next theorem states the asymptotic behavior of $\mathrm{HSIC}(P_n)$ under the alternative hypothesis $H_1 : X \not\perp\!\!\!\perp Y$.

**Theorem 3.3.** *Suppose that, for two multivariate random variables $X$ and $Y$, we have centered kernels $\tilde{k}_X$ and $\tilde{k}_Y$ with discrete eigenvalues. Suppose that Assumption 2.1 and Assumption 3.1 hold. If*

$$\text{there exists some } r, s \in \mathbb{N} \text{ such that } \mathbb{E}[\phi_{X,r}(X)\phi_{Y,s}(Y)] \neq 0, \tag{2}$$

*then*

$$n\,\mathrm{HSIC}(P_n) = \frac{1}{n}\,\mathrm{tr}(\widetilde{\boldsymbol{K}}_X \widetilde{\boldsymbol{K}}_Y) \xrightarrow{p} \infty \text{ as } m \to \infty.$$

*When $\tilde{k}_X$ and $\tilde{k}_Y$ are characteristic kernels, Condition (2) is equivalent to $H_1 : X \not\perp\!\!\!\perp Y$.*

Here Condition (2) is a sufficient condition for $X \not\perp\!\!\!\perp Y$: If $X \perp\!\!\!\perp Y$, then $\mathbb{E}[\phi_{X,r}(X)\phi_{Y,s}(Y)] = \mathbb{E}[\phi_{X,r}(X)]\,\mathbb{E}[\phi_{Y,s}(Y)] = 0$ for all $r, s \in \mathbb{N}$; as a contrapositive, (2) implies $X \not\perp\!\!\!\perp Y$. When characteristic kernels are used, based on the definition and property of the population HSIC, we can show that $X \not\perp\!\!\!\perp Y$ also implies (2).

The proof of Theorem 3.3 is provided in Appendix C. To prove the theorem, we show that, under Condition (2), there exists a statistic smaller than $d\,\mathrm{HSIC}(P_n)$ that converges in probability to a positive constant, which results in $n\,\mathrm{HSIC}(P_n) = md\,\mathrm{HSIC}(P_n)$ going to infinity, as the number of clusters ($m$) goes to infinity. When the test statistic goes to infinity, the rejection rate of the test would approach one. Hence, based on Theorem 3.3, we have established the consistency of the proposed test.

In practice, the weights $\ell_t$'s in (1) of Theorem 3.2 are unknown and we need to estimate them with empirical counterparts. In a similar spirit to Theorem 4 of Zhang et al. (2012) [22], the following proposition provides an approximation for the asymptotic null distribution of $\mathrm{HSIC}(P_n)$ and allows for independence testing in clustered data.

**Proposition 3.4.** *Assume that the conditions in Theorem 3.2 hold. To test the null hypothesis $H_0 : X \perp\!\!\!\perp Y$ at a significance level $\alpha$, we can compare the statistic $n\,\mathrm{HSIC}(P_n) = \frac{1}{n}\,\mathrm{tr}(\widetilde{\boldsymbol{K}}_X \widetilde{\boldsymbol{K}}_Y)$ against the $(1-\alpha)$-quantile of the distribution of*

$$\widetilde{T} = \frac{1}{m} \sum_{t=1}^{n^2} \tilde{\ell}_t z_t^2,$$

*where $z_t$'s are i.i.d. standard normal variables and $\tilde{\ell}_t$'s are eigenvalues of $\widetilde{\boldsymbol{V}}\widetilde{\boldsymbol{V}}^T$, with $\widetilde{\boldsymbol{V}} = [\tilde{\boldsymbol{v}}_1, \cdots, \tilde{\boldsymbol{v}}_m]$. Each vector $\tilde{\boldsymbol{v}}_i$ is obtained by vectorizing (i.e., stacking the columns of) the $n \times n$ matrix $\widetilde{\boldsymbol{M}}_i$, whose $(r,s)$-th entry is*

$$\widetilde{M}_{i,rs} = \frac{1}{\sqrt{d}} \sum_{j=di-d+1}^{di} \sqrt{\gamma_{X,r}\gamma_{Y,s}}\, u_{X,r}(X_j) u_{Y,s}(Y_j).$$

The proof of Proposition 3.4 is provided in Appendix D, where we show that $\widetilde{T}$ has the same asymptotic distribution as $n\,\mathrm{HSIC}(P_n)$ under $H_0$. Note that the eigenvalues of $\widetilde{\boldsymbol{V}}\widetilde{\boldsymbol{V}}^T$, an $n^2 \times n^2$ matrix, are the same as the eigenvalues of $\widetilde{\boldsymbol{V}}^T\widetilde{\boldsymbol{V}}$, an $m \times m$ matrix. In practice, we can calculate the eigenvalues of $\widetilde{\boldsymbol{V}}^T\widetilde{\boldsymbol{V}}$ instead to avoid excessive computational burden.

The distribution of $\widetilde{T}$, which is a mixture of chi-square variables, can be efficiently approximated by Davies' exact method [24]. This method is shown to work well in previous studies [1, 25] that have statistical tests based on a mixture of chi-square distributions.

# 4 Experiments

In this section, we conduct simulation studies and real data analysis to demonstrate the performance of our proposed test.

## 4.1 Simulation studies

### 4.1.1 Methods

We consider a longitudinal data setting, where a set of exposure variables $X \in \mathbb{R}^p$ and a set of outcome variables $Y \in \mathbb{R}^q$ are measured on $m$ subjects at 3 time points. In other words, the observations $\{(X_j, Y_j)\}_{j=1}^n$ are grouped into $m$ clusters with cluster size $d = 3$. To introduce correlation across different time points as well as across different variables, we use a Kronecker product-based covariance structure [7], which has often been used to model multivariate longitudinal data [11].

The general simulation setting is as following. For each cluster, let $x_{ij}$ denote the $i$-th variable in $X$ measured at the $j$-th time point, for $i = 1, \cdots, p$ and $j = 1, 2, 3$. Let $y_{ij}$ be defined similarly. Within each cluster, we let $(x_{11}, x_{12}, x_{13}, \cdots, x_{p1}, x_{p2}, x_{p3})^T \sim \mathcal{N}(5 \times \mathbf{1}_{3p}, \boldsymbol{\Sigma}_X)$, where $\boldsymbol{\Sigma}_X = \boldsymbol{R}_X \otimes \boldsymbol{R}_{cl}$, with

$$\boldsymbol{R}_X = \begin{pmatrix} 1 & \rho_X & \cdots & \rho_X \\ \rho_X & 1 & \cdots & \rho_X \\ \vdots & \vdots & \ddots & \vdots \\ \rho_X & \rho_X & \cdots & 1 \end{pmatrix}_{p \times p} , \quad \boldsymbol{R}_{cl} = \begin{pmatrix} 1 & \rho_c & \rho_c^2 \\ \rho_c & 1 & \rho_c \\ \rho_c^2 & \rho_c & 1 \end{pmatrix} .$$

Here $\otimes$ is the Kronecker product. Marginally, we have imposed an exchangeable correlation structure $\boldsymbol{R}_X$ across the $p$ variables in $X$ and an AR(1) correlation structure $\boldsymbol{R}_{cl}$ across the three time points. The correlations between distinct variables at different time points are products of the marginal correlations: e.g., $\mathbb{C}\mathrm{orr}(x_{11}, x_{22}) = \rho_X \rho_c$.

We simulate a situation where a single exposure (say, the $r$-th variable in $X$) affects multiple outcomes, with different effect sizes on different outcomes. Within each cluster, we use the model:

$$
\begin{aligned}
& (y_{11}, y_{12}, y_{13}, \cdots, y_{q1}, y_{q2}, y_{q3})^T \\
& = (\beta_1 f(x_{r1}), \beta_1 f(x_{r2}), \beta_1 f(x_{r3}), \cdots, \beta_q f(x_{r1}), \beta_q f(x_{r2}), \beta_q f(x_{r3}))^T + \boldsymbol{\epsilon},
\end{aligned}
\tag{3}
$$

where $\beta_s$, with $s = 1, \cdots, q$, is the effect size of the chosen exposure on the $s$-th outcome, and $\boldsymbol{\epsilon} \sim \mathcal{N}(0, \boldsymbol{\Sigma}_Y)$, with $\boldsymbol{\Sigma}_Y = \boldsymbol{R}_Y \otimes \boldsymbol{R}_{cl}$. $\boldsymbol{R}_Y$ is the correlation matrix for the $q$ variables in $Y$. In simulations, we set $p = q = 20$, $\rho_X = 0.5$ and consider various levels of within-cluster correlation: $\rho_c = 0.3, 0.5$ or $0.7$. We also let $\boldsymbol{R}_Y = \boldsymbol{R}_X$.

**Type I error simulation** To evaluate the type I error rate (rejection rate under $H_0$), we let $\beta_1 = \cdots = \beta_q = 0$, so that the null hypothesis $H_0 : X \perp\!\!\!\perp Y$ is true. We perform both the proposed HSIC test with proper accommodation for clustered correlation (**HSIC$_{\mathbf{cl}}$**), and the original HSIC test without any adjustment (**HSIC$_{\mathbf{orig}}$**) as in [22] and [1]. These two methods are applied to the data $\{(X_j, Y_j)\}_{j=1}^n$ at the observation level. We consider $m = 500, 1000$ or $1500$ clusters and calculate the empirical type I error rates in each setting based on 1000 simulated data sets.

From Model (3), both $X$ and $Y$ have multivariate normal distributions under $H_0$. While we focus on normal data here, additional Type I error simulations based on non-normal data are considered in Appendix F.1.

**Power simulation** To evaluate the power (rejection rate under $H_1$), we randomly select one exposure variable from $X$ to be the causal exposure, and make the first $\eta$ proportion ($\eta = 10\%, 20\%, 30\%, 40\%$) of outcomes in $Y$ depend on that exposure (with nonzero $\beta_s$'s). We let the function $f(x)$ take two forms: $f(x) = x$ (**Power Scenario 1**) and $f(x) = \log((x - 4)^2)$ (**Power**

**Scenario 2**). For $s = 1, \cdots, \eta q$, the effect sizes $\beta_s$'s are generated from a Uniform$(0, \sqrt{25/m})$ distribution under **Power Scenario 1**, and from Uniform$(0, \sqrt{10/m})$ under **Power Scenario 2**.

In the power simulation, we perform $\mathbf{HSIC_{cl}}$ and two other HSIC-based competing methods. The two competing methods analyze data at the cluster/subject level. In the first method ($\mathbf{HSIC_{mean}}$), for each cluster, we take an average of observations at different time points: We consider the new variables $X^* := (\frac{1}{3}\sum_{j=1}^3 x_{1j}, \cdots, \frac{1}{3}\sum_{j=1}^3 x_{pj})^T$ and $Y^* := (\frac{1}{3}\sum_{j=1}^3 y_{1j}, \cdots, \frac{1}{3}\sum_{j=1}^3 y_{qj})^T$ and then perform the original HSIC test based on $\{(X_i^*, Y_i^*)\}_{i=1}^m$. In the second method ($\mathbf{HSIC_{cat}}$), we follow the strategy of Rudra et al. [19] and concatenate the observations at different time points for each cluster: We consider the new variables $X^{**} := (x_{11}, x_{12}, x_{13}, \cdots, x_{p1}, x_{p2}, x_{p3})^T$ and $Y^{**} := (y_{11}, y_{12}, y_{13}, \cdots, y_{q1}, y_{q2}, y_{q3})^T$ and then perform the original HSIC test based on $\{(X_i^{**}, Y_i^{**})\}_{i=1}^m$.

We consider $m = 500$ clusters and calculate the empirical power in each setting based on 1000 simulated data sets.

**Kernel choices** For both $X$ and $Y$, we consider two different kernels: the Gaussian kernel $k_X(z_1, z_2) = k_Y(z_1, z_2) = \exp(-\|z_1 - z_2\|_2^2/\tau)$ and the linear kernel $k_X(z_1, z_2) = k_Y(z_1, z_2) = z_1^T z_2$. For the Gaussian kernel, the shape parameter $\tau$ is chosen as the median of the Euclidean distance between each sample pair. While the Gaussian kernel is a characteristic kernel [21], the linear kernel is not characteristic and is designed to detect linear or close-to-linear relationships between $X$ and $Y$. Nevertheless, linear kernels have been shown to be reasonably powerful in previous association studies [25, 2] and can be computationally efficient (see Appendix G.2).

Additional simulation studies are provided in Appendix F. Additional implementation details including computation time and code availability are provided in Appendix G.

### 4.1.2 Results

Table 1 shows the empirical type I error rates of $\mathbf{HSIC_{orig}}$ and $\mathbf{HSIC_{cl}}$ for normal data. The type I error rate of $\mathbf{HSIC_{orig}}$ is inflated in each setting, where the inflation becomes greater as the within-cluster correlation ($\rho_c$) increases. In contrast, $\mathbf{HSIC_{cl}}$ has a well-controlled type I error rate across all levels of within-cluster correlation. Using the linear kernel, $\mathbf{HSIC_{cl}}$ has type I error rates close to the nominal $\alpha$ in all situations. Using the Gaussian kernel, $\mathbf{HSIC_{cl}}$ is conservative when the number of clusters is moderate ($m = 500$), but its type I error rate gets close to the nominal $\alpha$ at a larger sample size ($m = 1500$). For non-normal data (Figure F1 in Appendix), the pattern is similar: $\mathbf{HSIC_{cl}}$ is able to control the type I error rate, either with the Gaussian kernel or with the linear kernel.

$\mathbf{HSIC_{cl}}$ based on the Gaussian kernel is more conservative, likely because the Gaussian kernel is associated with a larger number of non-zero eigenvalues in finite samples than the linear kernel in our simulation setting. The null distribution for Gaussian-kernel-based $\mathbf{HSIC_{cl}}$ thus involves more terms in the weighted sum of chi-square variables (as the weights depend on eigenvalues), which might aggregate more finite-sample errors and make the test statistic converge slower to the asymptotic distribution.

Figure 1 shows the empirical power of $\mathbf{HSIC_{cl}}$ and the two competing methods under **Power Scenario 1**. In all situations, $\mathbf{HSIC_{cl}}$ has a higher power than both $\mathbf{HSIC_{mean}}$ and $\mathbf{HSIC_{cat}}$, regardless of the level of within-cluster correlation or the kernel being used. In addition, the power of $\mathbf{HSIC_{cl}}$ improves quickly as a higher proportion of variables in $Y$ is associated with $X$. Since $X$ and $Y$ are linearly associated in Scenario 1, it is not surprising that the linear kernel is powerful in detecting this dependence. The Gaussian kernel has a comparable performance as the linear kernel.

Figure 2 shows the empirical power under **Power Scenario 2**, where $X$ and $Y$ have a non-linear relationship. Similar to **Power Scenario 1**, for both the Gaussian kernel and the linear kernel, $\mathbf{HSIC_{cl}}$ achieves a higher power than the competing methods under all levels of within-cluster correlation. When compared between kernels, $\mathbf{HSIC_{cl}}$ based on the Gaussian kernel is more powerful than $\mathbf{HSIC_{cl}}$ based on the linear kernel in each setting, showing the advantage of the Gaussian kernel as a characteristic kernel to detect general dependence patterns.

Overall, both **Power Scenario 1** and **2** show the considerable power gain of $\mathbf{HSIC_{cl}}$ over analyzing data at the cluster level. We also note that, the power gain of $\mathbf{HSIC_{cl}}$ decreases as the within-cluster correlation increases. This is expected since there will be less pronounced information loss in

Table 1: Empirical type I error rate of **HSIC_orig** and **HSIC_cl** at nominal level $\alpha$ for normal data under simulation.

| $\alpha$ | $m$ | $\rho_c$ | Gaussian kernel | | Linear kernel | |
|---|---|---|---|---|---|---|
| | | | **HSIC_orig** | **HSIC_cl** | **HSIC_orig** | **HSIC_cl** |
| 0.05 | 500 | 0.3 | 0.119 | 0.024 | 0.068 | 0.047 |
| | | 0.5 | 0.603 | 0.031 | 0.141 | 0.044 |
| | | 0.7 | 1.000 | 0.030 | 0.330 | 0.044 |
| | 1000 | 0.3 | 0.115 | 0.029 | 0.070 | 0.043 |
| | | 0.5 | 0.591 | 0.034 | 0.166 | 0.054 |
| | | 0.7 | 1.000 | 0.034 | 0.348 | 0.043 |
| | 1500 | 0.3 | 0.113 | 0.047 | 0.082 | 0.053 |
| | | 0.5 | 0.608 | 0.043 | 0.145 | 0.053 |
| | | 0.7 | 1.000 | 0.044 | 0.352 | 0.052 |
| 0.01 | 500 | 0.3 | 0.018 | 0.005 | 0.021 | 0.013 |
| | | 0.5 | 0.190 | 0.005 | 0.035 | 0.011 |
| | | 0.7 | 0.998 | 0.009 | 0.114 | 0.008 |
| | 1000 | 0.3 | 0.022 | 0.006 | 0.015 | 0.010 |
| | | 0.5 | 0.180 | 0.010 | 0.050 | 0.008 |
| | | 0.7 | 0.999 | 0.010 | 0.111 | 0.010 |
| | 1500 | 0.3 | 0.027 | 0.007 | 0.019 | 0.010 |
| | | 0.5 | 0.209 | 0.008 | 0.047 | 0.010 |
| | | 0.7 | 0.999 | 0.008 | 0.117 | 0.009 |

averaging or concatenating the data at the cluster level if observations within a cluster are highly correlated.

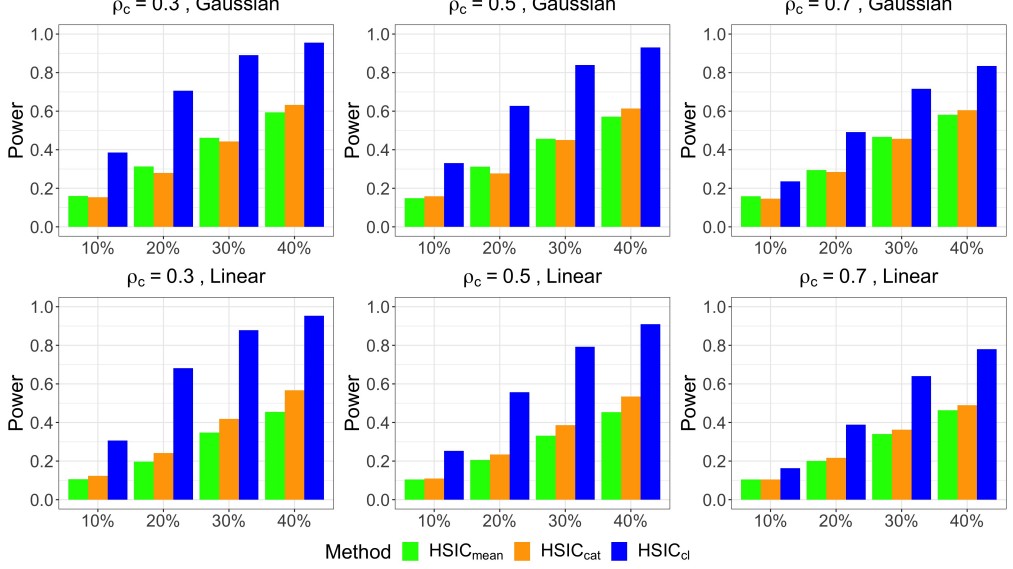

Figure 1: Empirical power of **HSIC_cl** and competing methods at nominal level $\alpha = 0.05$ under **Power Scenario 1**. The x-axis represents the proportion of variables in $Y$ that are associated with $X$. The top row shows results based on the Gaussian kernel, and the bottom row shows results based on the linear kernel.

While the above results are based on a fixed cluster size, we have also investigated the effect of cluster size on the performance of **HSIC_cl** (Appendix F.2). When the number of clusters ($m$) and the level of within-cluster correlation ($\rho_c$) are fixed, type I error control is similar for different cluster sizes (Figure F2-F3), suggesting that the convergence speed of the test statistic under $H_0$ is likely not affected by cluster size. However, a larger cluster size tends to result in a higher statistical

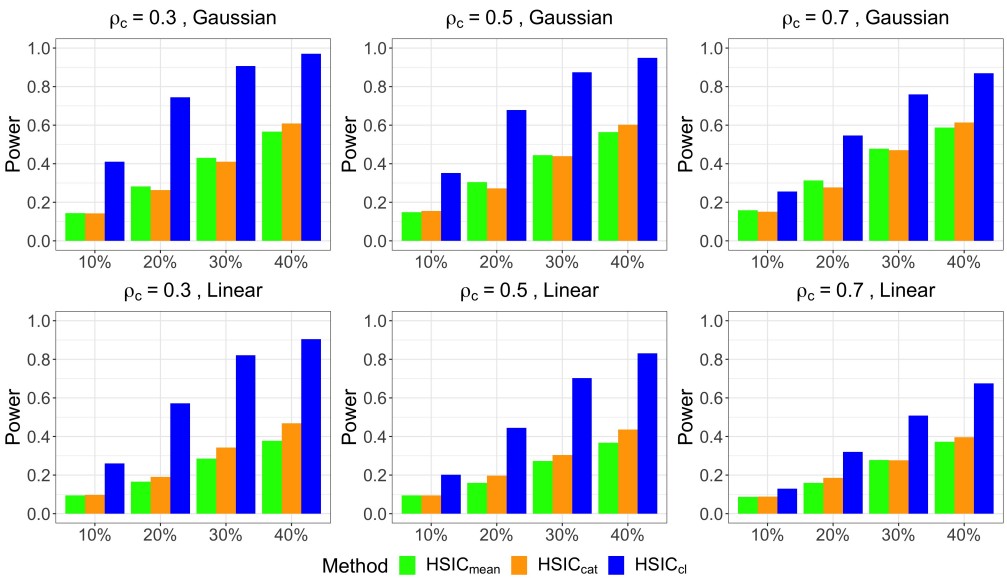

Figure 2: Empirical power of $\mathbf{HSIC_{cl}}$ and competing methods at nominal level $\alpha = 0.05$ under **Power Scenario 2**. The x-axis represents the proportion of variables in $Y$ that are associated with $X$. The top row shows results based on the Gaussian kernel, and the bottom row shows results based on the linear kernel.

power (Figure F4), possibly due to an increase in the overall sample size, which allows for additional information gain.

## 4.2 Application to real data

The vaginal microbiota plays an important role in maintaining vaginal homeostasis. Common vaginal conditions, such as bacterial vaginosis, are characterized by shifts in vaginal microbiome composition and changes in vaginal metabolites [4, 26]. Studying the association between the microbiome and the metabolites helps us better understand how the vaginal microbiota contributes to the host metabolic environment, and identifies potential metabolic biomarkers for vaginal conditions [4]. Here we apply $\mathbf{HSIC_{cl}}$ and competing methods to test the dependence between the overall vaginal microbiome composition and different metabolic pathways, using data from the Menopause Strategies: Finding Lasting Answers for Symptoms and Health (MsFLASH) Vaginal Health Trial [27].

The MsFLASH trial was a 12-week randomized clinical trial to evaluate the treatment effect of vaginal estradiol vs. placebo on vaginal discomfort in postmenopausal women [27] (see Appendix H.1 for more details). As part of an effort to investigate the mechanism of postmenopausal vaginal discomfort, vaginal microbiota and vaginal fluid metabolites were characterized longitudinally and available in 141 participants at baseline, 4 and 12 weeks [28]. The vaginal microbiome profiles included abundance data of 381 taxa. The metabolome profiles included abundance data of 171 metabolites, which were grouped into 95 metabolic pathways. We apply $\mathbf{HSIC_{cl}}$ , $\mathbf{HSIC_{mean}}$ and $\mathbf{HSIC_{cat}}$ to assess the dependence between the overall vaginal microbiome composition and the metabolites in each pathway, across all 95 pathways. In other words, for each test, we have $m = 141$, $d = 3$, $X \in \mathbb{R}^{381}$ and $Y \in \mathbb{R}^q$, where $q$ is the number of metabolites in a pathway, ranging from 1 to 21 in this data set; 95 tests are performed in total.

Table 2 shows the number of metabolic pathways identified to be associated with the vaginal microbiome composition, at a Bonferroni-corrected significance level $\alpha = 0.05/95 = 5.3 \times 10^{-4}$. Due to the close relationship between vaginal microbiota and vaginal metabolites, all methods have identified a considerable number of significant metabolic pathways. Still, $\mathbf{HSIC_{cl}}$ identifies a larger number of pathways than $\mathbf{HSIC_{mean}}$ and $\mathbf{HSIC_{cat}}$, either with the Gaussian kernel or with the linear kernel. In particular, based on the Gaussian kernel, $\mathbf{HSIC_{cl}}$ successfully identifies all the significant pathways discovered by $\mathbf{HSIC_{mean}}$ and $\mathbf{HSIC_{cat}}$, and discovers 4 (7) additional pathways compared to $\mathbf{HSIC_{cat}}$ ($\mathbf{HSIC_{mean}}$) (Figure H1). 67 out of 68 pathways discovered by $\mathbf{HSIC_{cl}}$ using

the linear kernel are also identified by **HSICcl** using the Gaussian kernel (Figure H2). For this data set, the Gaussian kernel appears to be more powerful in detecting dependence than the linear kernel, indicating a possibly non-linear relationship between certain metabolites and microbial taxa abundances.

Table 2: Number of metabolic pathways identified to be associated with the vaginal microbiome composition based on the MsFLASH data set ($\alpha = 5.3 \times 10^{-4}$).

| Kernel | $\mathbf{HSIC_{mean}}$ | $\mathbf{HSIC_{cat}}$ | $\mathbf{HSIC_{cl}}$ |
|---|---|---|---|
| Gaussian | 68 | 71 | 75 |
| Linear | 64 | 64 | 68 |

We focus on some of the top pathways (with high statistical significance) identified using $\mathbf{HSIC_{cl}}$ and highlight their biological relevance. The top pathways include multiple metabolic pathways for amino acids. The human vaginal microbiota is dominated by bacteria in the *Lactobacillus* genus [29], which are known to produce branched-chain amino acids including valine, leucine and isoleucine [30]. All these amino acids are present in our top pathways. In particular, one pathway related to leucine metabolism is only identified by $\mathbf{HSIC_{cl}}$ but not by $\mathbf{HSIC_{mean}}$ or $\mathbf{HSIC_{cat}}$. Therefore, our finding is consistent with previous studies on bacterial metabolism, confirming the power improvement in using $\mathbf{HSIC_{cl}}$ for scientific discovery.

## 5 Discussion

We have introduced a novel kernel-based approach, $\mathbf{HSIC_{cl}}$, to evaluate the generalized dependence between two multivariate variables based on cluster-correlated data. Using the previously developed HSIC statistic as our test statistic, we have derived its asymptotic null distribution in the presence of clustered correlation and constructed a statistical test of independence accordingly. We have also established the consistency of the proposed test. Both simulation studies and application to real longitudinal data demonstrate the power gain in using our proposed test, compared to methods based on measurements averaged or concatenated at the cluster level.

One limitation of our framework is that the proposed test only applies to balanced and complete clustered data, which might not be always available in practice. In longitudinal studies, for example, subjects might be followed at different time points from one another (resulting in unbalanced data), or become lost to follow-up (resulting in incomplete data). For incomplete data, one solution is to impute the missing data before applying $\mathbf{HSIC_{cl}}$. Further extension of the test for unbalanced or incomplete clustered data will be interesting for future study.

Another limitation is that our proposed test relies on asymptotic results, and the null distribution might not be accurately approximated when the number of clusters is small, which is likely true of many family-based or longitudinal studies. Permutation-based approaches could be a surrogate for $\mathbf{HSIC_{cl}}$ at small sample sizes (see Appendix F.3), although their computational burden is large compared to $\mathbf{HSIC_{cl}}$ (see Table G1). Computationally efficient adaptations of $\mathbf{HSIC_{cl}}$ to small sample sizes, such as those proposed by Lee et al. [31] and Zhan et al. [32], would be another useful extension.

With the continuing emergence of high-dimensional data and the prevalence of cluster-correlated data in different scientific fields, our proposed test is a promising approach to discover novel associations and bring new scientific insights in these settings. Meanwhile, we need to be cautious about potential risks to society that might result from misuse or misinterpretation of our proposed test. For example, confounding is an important factor to consider in genetic and epidemiological studies. A confounder affects both the exposure and the outcome, and could lead to spurious associations between the two variables even if the variables themselves do not have causal relationships. Therefore, as one applies our proposed test to evaluate the association between two variables, it is important to consider the presence of potential confounders and be careful in interpreting the test results. Mistaking certain observed correlation for causation could lead to misinformation in the scientific community and would be especially concerning when the studies being conducted directly influence people's life.

## Acknowledgments and Disclosure of Funding

This work is supported by the National Institutes of Health grant R01-GM129512. The MsFLASH data were provided by the Fred Hutchinson Cancer Research Center (MsFLASH Network), which is supported by the National Institute on Aging grant 5R01-AG048209. We would like to thank the anonymous reviewers for their helpful feedback.

The authors declare no competing interests.

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
