# Appendix for: A Kernel-based Test of Independence for Cluster-correlated Data

## Contents

## A   Preliminary results

In this section, we present some preliminary results that will be useful in proving Theorem 3.2, Theorem 3.3 and Proposition 3.4. We draw upon existing theory on properties of random kernel matrices and extend these properties to cluster-correlated data. Specifically, we show the convergence of eigenvalues and eigenvectors of an empirical kernel matrix based on clustered data.

Let $(\mathcal{X}, \mathcal{F}, P)$ be a probability space and $\mathcal{H}$ be a Hilbert space over $(\mathcal{X}, \mathcal{F}, P)$ with a symmetric kernel function $k : \mathcal{X} \times \mathcal{X} \to \mathbb{R}$. Let $H$ be a compact operator on $\mathcal{H}$, defined by

$$Hg(x) = \int_{\mathcal{X}} k(x, x')g(x')dP(x') \quad \text{for } x \in \mathcal{X}, g \in \mathcal{H}.$$

Let $\mathcal{H}_n$ be the Hilbert space over $(\mathcal{X}, \mathcal{F}, P_n)$, where $P_n = \frac{1}{n} \sum_{j=1}^n \delta_{X_j}$ is the empirical version of $P$ for a given $n \in \mathbb{N}$. Let $H_n : \mathcal{H}_n \to \mathcal{H}_n$ be the empirical version of the operator $H$, defined by

$$H_n g(x) = \int_{\mathcal{X}} k(x, x') g(x') dP_n(x') \quad \text{for } x \in \mathcal{X}, g \in \mathcal{H}_n.$$

Equivalently, $H_n$ can be viewed as an $n \times n$ real matrix whose $(i, j)$-th entry is

$$\{H_n\}_{i,j} = \frac{1}{n} k(X_i, X_j).$$

This is the empirical kernel matrix scaled by a factor of $1/n$.

Here we restrict our discussion to a reproducing kernel Hilbert space (RKHS) $\mathcal{H}$, where the kernel function $k$ is positive semi-definite. We also assume that the operator $H$ is Hilbert–Schmidt, with $\mathbb{E}[k^2(X, X')] < \infty$.

Let $\lambda(T)$ denote the spectrum of a compact, symmetric operator $T$. Then $\lambda(H)$ and $\lambda(H_n)$ are the sets of eigenvalues for $H$ and $H_n$, respectively. Since $H_n$ is an operator on $\mathbb{R}^n$, we add to its spectrum an infinite number of zeros, such that $\lambda(H)$ and $\lambda(H_n)$ are comparable. For an operator with a positive semi-definite kernel, the associated eigenvalues are non-negative.

For any two compact, symmetric operators $A$ and $B$ with positive semi-definite kernels, let $a_1 \geq a_2 \geq \cdots \geq 0$ be the eigenvalues in $\lambda(A)$ arranged in a non-increasing order and let $b_1 \geq b_2 \geq \cdots \geq 0$ be the eigenvalues in $\lambda(B)$ arranged in a non-increasing order. Following the work by Koltchinskii et al. [1], we can define a distance measure $\delta_2$ on $\ell_2(\mathbb{N})$ such that

$$\delta_2(\lambda(A), \lambda(B)) = \left[ \sum_i (a_i - b_i)^2 \right]^{1/2}.$$

As shown in [1], the measure $\delta_2$ is a well-defined distance between spectra of Hilbert–Schmidt operators or operators on $\mathbb{R}^n$. It satisfies the triangle inequality with

$$\delta_2(\lambda(A), \lambda(B)) \leq \delta_2(\lambda(A), \lambda(C)) + \delta_2(\lambda(C), \lambda(B))$$

for any operators $A$, $B$ and $C$.

We now consider a sample $\underline{X}_n = (X_1, \cdots, X_n)$ with clustered correlation among the observations, as defined in Section 2.1 of the main text.

**Assumption A.1.** *Assume that $\underline{X}_n$ can be divided into $m$ i.i.d. clusters of fixed size $d$ (i.e., $n = md$). The observations $X_1, \cdots, X_n$ are identically distributed according to $P$, and the clusters $\{[X_{di-d+1}, X_{di-d+2}, \cdots, X_{di}]\}_{i=1}^m$ are independent from each other while having identical within-cluster correlation structure.*

## A.1  Convergence of eigenvalues

We show that, with clustered data, the set of eigenvalues for $H_n$ converges to the set of eigenvalues for $H$ as the number of clusters goes to infinity. We first introduce a lemma that will be useful in proving this statement.

**Lemma A.2.** *Suppose that Assumption A.1 holds for the sample $\underline{X}_n$. Let $V_n$ be a V-statistic based on $\underline{X}_n$ with a bivariate symmetric kernel function $f(x, x')$:*

$$V_n := \frac{1}{n^2} \sum_{i=1}^n \sum_{j=1}^n f(X_i, X_j).$$

*Assume that $V_n$ is non-degenerate and the class $\mathcal{C} := \{x \mapsto f(x, x') : x' \in \mathcal{X}\}$ is a P-Donsker class [2], then*

$$V_n \xrightarrow{p} \mathbb{E}[f(X, X')] \text{ as } m \to \infty,$$

*where $X'$ is an independent copy of $X$.*

The proof of Lemma A.2 is provided in Appendix E.

We have assumed that the operator $H$ is Hilbert–Schmidt and the kernel $k$ is positive semi-definite. By Mercer's Theorem (see, e.g. Theorem 3.6 of [3]), there exists an orthonormal set $\{\phi_r : r \in J\}$

of $\mathcal{H}$, where $J \subseteq \mathbb{N}$, and a sequence of descending non-negative real numbers $\{\lambda_r : r \in J\}$ with $\sum_{r \in J} \lambda_r^2 < \infty$ such that

$$k(x, x') = \sum_{r \in J} \lambda_r \phi_r(x) \phi_r(x').$$

Here the set $\lambda(H) := \{\lambda_r : r \in J\}$ is the set of eigenvalues for $H$, and the set $\{\phi_r : r \in J\}$ is the corresponding set of eigenfunctions. For any fixed $R \in \mathbb{N}$, we further define $k_R(x, x') := \sum_{r=1}^{R} \lambda_r \phi_r(x) \phi_r(x')$.

The following theorem states the convergence of eigenvalues of the empirical kernel matrix. The theorem and associated proof are adapted from Theorem 3.1 of Koltchinskii et al. [1], where we extend the assumption from i.i.d. data to clustered data.

**Theorem A.3.** *Assume that the kernel $k(\cdot, \cdot)$ is symmetric and positive semi-definite, and $\mathbb{E}[k^2(X, X')] < \infty$. Suppose that Assumption A.1 holds for the sample $\underline{X}_n$, and $\mathcal{C} := \{x \mapsto (k - k_R)^2(x, x') : x' \in \mathcal{X}\}$ is a P-Donsker class for any $R \in \mathbb{N}$. Then we have*

$$\delta_2(\lambda(H_n), \lambda(H)) \xrightarrow{P} 0 \text{ as } m \to \infty. \tag{1}$$

*As a result, the $r$-th largest eigenvalue of $H_n$ converges in probability to the $r$-th largest eigenvalue of $H$, for each $r$.*

*Proof.* Symmetric kernel $k$ and $\mathbb{E}[k^2(X, X')] < \infty$ ensure that the operator $H$ is Hilbert–Schmidt. For the decomposition $k(x, x') = \sum_{r \in J} \lambda_r \phi_r(x) \phi_r(x')$, we consider first the basic case where $J = \{1, \cdots, R_0\}$ for some $R_0 < \infty$, and then the general case where $J = \mathbb{N}$.

In the basic case, we have $k(x, x') = \sum_{r=1}^{R_0} \lambda_r \phi_r(x) \phi_r(x')$. Following the argument of Koltchinskii et al. [1], we can prove that (1) holds, using tools of operator perturbation theory. The majority of the proof in the basic case would be similar to that of Koltchinskii et al. and we skip the details here. To account for clustered data, in Eq. (3.4) of [1], we apply the law of large numbers (LLN) to the clusters instead of individual observations.

We next focus on the general case where $J = \mathbb{N}$. For any fixed $R < \infty$, let $H_R$ be the integral operator with kernel $k_R(x, x') = \sum_{r=1}^{R} \lambda_r \phi_r(x) \phi_r(x')$. Then we have

$$\lim_{R \to \infty} \delta_2(\lambda(H), \lambda(H_R)) = \lim_{R \to \infty} \left[ \sum_{r=R+1}^{\infty} \lambda_r^2 \right]^{1/2} = 0. \tag{2}$$

Let $H_{R,n}$ be an $n \times n$ real matrix (i.e., an operator on $\mathbb{R}^n$) whose $(i, j)$-th entry is

$$\{H_{R,n}\}_{i,j} = \frac{1}{n} k_R(X_i, X_j).$$

By the result (1) established in the basic case, we have

$$\lim_{m \to \infty} \delta_2(\lambda(H_{R,n}), \lambda(H_R)) = 0 \text{ in probability for all } R < \infty. \tag{3}$$

Now, by Hoffman–Wielandt Inequality (Theorem 2.2 of [1]), we have

$$\lim_{R \to \infty} \lim_{m \to \infty} \delta_2(\lambda(H_{R,n}), \lambda(H_n)) \leq \lim_{R \to \infty} \lim_{m \to \infty} \|H_{R,n} - H_n\|_{HS}$$
$$= \lim_{R \to \infty} \lim_{m \to \infty} \left[ \frac{1}{n^2} \sum_{1 \leq i,j \leq n} (k - k_R)^2(X_i, X_j) \right]^{1/2}. \tag{4}$$

Here $V_n := \frac{1}{n^2} \sum_{1 \leq i,j \leq n} (k - k_R)^2(X_i, X_j)$ is a V-statistic with kernel $f(x, x') = (k - k_R)^2(x, x')$. By Lemma A.2, which shows the convergence in probability of a V-statistic based on clustered data, we have

$$V_n \xrightarrow{P} \mathbb{E}[(k - k_R)^2(X, X')] \text{ as } m \to \infty,$$

where $X'$ is an independent copy of $X$. Therefore, (4) becomes

$$\lim_{R \to \infty} \lim_{m \to \infty} \delta_2(\lambda(H_{R,n}), \lambda(H_n)) \leq \lim_{R \to \infty} \mathbb{E}[(k - k_R)^2(X, X')]^{1/2}$$
$$= \lim_{R \to \infty} \left[ \sum_{r=R+1}^{\infty} \lambda_r^2 \right]^{1/2} = 0 \text{ in probability.} \tag{5}$$

Finally, combining (2), (3) and (5), we have

$$\lim_{m \to \infty} \delta_2(\lambda(H_n), \lambda(H))$$

$$\leq \lim_{R \to \infty} \limsup_{m \to \infty} \Big[ \delta_2(\lambda(H_n), \lambda(H_{R,n})) + \delta_2(\lambda(H_{R,n}), \lambda(H_R)) + \delta_2(\lambda(H_R), \lambda(H)) \Big]$$

$$= 0 \text{ in probability.}$$

Suppose $\gamma_r$ is the $r$-th largest eigenvalue of $H_n$, and recall that $\lambda_r$ is the $r$-th largest eigenvalue of $H$. For any $r \in \mathbb{N}$, we have that

$$|\gamma_r - \lambda_r| \leq \Big[ \sum_i (\gamma_i - \lambda_i)^2 \Big]^{1/2} = \delta_2(\lambda(H_n), \lambda(H)) \xrightarrow{p} 0 \text{ as } m \to \infty.$$

We have thus proved the theorem. $\qquad\qquad\qquad\qquad\qquad\qquad\qquad\qquad\qquad\qquad$ $\square$

We now show that some common kernel functions such as linear kernels and Gaussian kernels satisfy the conditions in Theorem A.3.

For any $\mathcal{X} \subseteq \mathbb{R}^p$, the linear kernel $k(x, x') = x^T x'$ has exactly $p$ non-zero eigenvalues. Therefore, the linear kernel trivially satisfies the basic case in the proof of Theorem A.3.

Next, we show that conditions in Theorem A.3 hold for a Gaussian kernel $k(x, x') = \exp(-\|x - x'\|_2^2/(2\sigma^2))$ under $\mathcal{X} = \mathbb{R}$ and a normal distribution $P$. The case where $\mathcal{X}$ is multi-dimensional can be generalized from the univariate case.

**Proposition A.4.** *Suppose that $\mathcal{X} = \mathbb{R}$ and $P = \mathcal{N}(0, \tau^2)$. Given a Gaussian kernel $k(x, x') = \exp(-(x - x')^2/(2\sigma^2))$, it holds that the class $\mathcal{C} := \{x \mapsto (k - k_R)^2(x, x') : x' \in \mathcal{X}\}$ is a P-Donsker class.*

*Proof.* We utilize the fact that, classes of functions with uniformly bounded variation are $P$-Donsker [4]. We will show that there exists some $M < \infty$ such that the total variation norm $\|\cdot\|_V$ of each function $f$ in $\mathcal{C}$ satisfies: $\|f\|_V \leq M$. The total variation norm of a differentiable function $f$ is of the form: $\|f\|_V = \int |f'(x)| dx$.

As shown in Section 4.3.1 of [5], the eigenvalues $\lambda_r$ and eigenfunctions $\phi_r$ of the Gaussian kernel $k(x, x') = \exp(-(x - x')^2/(2\sigma^2))$, with $r = 0, 1, 2, \cdots$, are of the following form:

$$\lambda_r = \sqrt{\frac{2a}{A}} B^k, \quad \phi_r(x) = \exp(-(c - a)x^2) H_r(\sqrt{2c}x),$$

where $H_r(x) = (-1)^r \exp(x^2) \frac{d^r}{dx^r} \exp(-x^2)$ is the $r$-th order Hermite polynomial, $a = 1/(4\tau^2)$, $b = 1/(2\sigma^2)$ and

$$c = \sqrt{a^2 + 2ab}, \quad A = a + b + c, \quad B = b/A.$$

Therefore, we can express $k_R(x, x') = \sum_{r=0}^{R-1} \lambda_r \phi_r(x) \phi_r(x')$ in a closed form.

Both $k$ and $k_R$ are differentiable. Fixing $x' \in \mathcal{X}$, by definition of total variation norm, we have

$$\|(k - k_R)^2(\cdot, x')\|_V = \int \Big| \frac{d}{dx} (k - k_R)^2(x, x') \Big| dx$$

$$= \int \Big| 2(k - k_R)(x, x') \Big[ -\frac{1}{\sigma^2}(x - x') k(x, x') - k_R^*(x, x') \Big] \Big| dx,$$

where $k_R^*(x, x') = \sum_{r=0}^{R-1} \lambda_r \phi_r^*(x) \phi_r(x')$, with

$$\phi_r^*(x) = \begin{cases} \exp(-(c - a)x^2)\big[ -2(c - a)x \big] & \text{if } r = 0, \\ \exp(-(c - a)x^2)\big[ 2r\sqrt{2c}H_{r-1}(\sqrt{2c}x) - 2(c - a)x H_r(\sqrt{2c}x) \big] & \text{if } r > 0. \end{cases}$$

By triangle inequality and Cauchy-Schwarz inequality, $\|(k - k_R)^2(\cdot, x')\|_V$ is uniformly bounded (with respect to $x'$) as long as the following terms are uniformly bounded:

$$\int (x - x')^2 k^2(x, x') dx, \quad \int k^2(x, x') dx, \quad \int k_R^2(x, x') dx, \quad \int (k_R^*(x, x'))^2 dx.$$

We have:

$$\int (x-x')^2 k^2(x,x')dx = \int (x-x')^2 \exp\Big(-\frac{(x-x')^2}{\sigma^2}\Big)dx = \sqrt{\pi}\sigma\, \mathbb{E}_{P_0}[(X-x')^2] = \frac{\sqrt{\pi}\sigma^3}{2},$$

where the expectation is evaluated according to the distribution $P_0 = \mathcal{N}(x', \sigma^2/2)$. Similarly, it is easy to show that $\int k^2(x,x')dx = \sqrt{\pi}\sigma$.

To show that $\int k_R^2(x,x')dx$ is uniformly bounded, by Cauchy-Schwarz inequality, it suffices to show that $\int \phi_r^2(x)dx$ and $\phi_r(x')\phi_s(x')$ are uniformly bounded for any $r,s \in \{0,\cdots,R-1\}$ and $x' \in \mathcal{X}$. We have

$$\int \phi_r^2(x)dx = \int \exp(-2(c-a)x^2)H_r^2(\sqrt{2c}x)dx = \sqrt{\frac{\pi}{2(c-a)}}\, \mathbb{E}_{P_1}[H_r^2(\sqrt{2c}X)],$$

where the expectation is evaluated according to the distribution $P_1 = \mathcal{N}(0, 1/[4(c-a)])$. The $r$-th order Hermite polynomial is a polynomial of degree $r$. Since a normal distribution has finite $r$-th moments for any non-negative integer $r$, the term $\mathbb{E}_{P_1}[H_r^2(\sqrt{2c}X)]$ is finite.

Let $M_1 = \max_{r\in\{0,\cdots,R-1\}} \mathbb{E}_{P_1}[H_r^2(\sqrt{2c}X)]$, then $\int \phi_r^2(x)dx \leq M_1$ for all $r \in \{0,\cdots,R-1\}$.

Note that $\phi_r(x') = \exp(-(c-a)(x')^2)H_r(\sqrt{2c}x')$ is a bounded function for any $r$: Intuitively, the exponential part changes at a larger rate than the polynomial part, and $\exp(-(c-a)(x')^2)$ is bounded between 0 and 1. Therefore, there exists $c_0,\cdots,c_{R-1} > 0$ such that $\sup_{x'\in\mathcal{X}} |\phi_r(x')| \leq c_r$ for each $r = 0,\cdots,R-1$.

Let $M_2 = \max_{r,s\in\{0,\cdots,R-1\}} c_r c_s$, then $|\phi_r(x')\phi_s(x')| \leq M_2$ for all $r,s \in \{0,\cdots,R-1\}$ and $x' \in \mathcal{X}$. Similarly, we can show that $\int (k_R^*(x,x'))^2 dx$ is also uniformly bounded.

As a result, for each $\sigma^2$ and $\tau^2$, there exists $M < \infty$ such that, for any $x' \in \mathcal{X}$, the function $f(x) = (k-k_R)^2(x,x')$ has a total variation bounded by $M$. Thus $\mathcal{C} = \{x \mapsto (k-k_R)^2(x,x') : x' \in \mathcal{X}\}$ is a $P$-Donsker class. $\qquad \square$

## A.2 Convergence of eigenvectors

We next show that, under sufficient conditions, the eigenvectors of the scaled empirical kernel matrix $H_n$ would converge in probability to the corresponding eigenfunctions for the operator $H$, as the number of clusters goes to infinity. In particular, the $i$-th element of the $r$-th eigenvector of $H_n$ converges in probability to the $r$-th eigenfunction of $H$ evaluated at $X_i$, up to some scaling.

Let $\lambda(H) := \{\lambda_r : r \in \mathbb{N}\}$ be the set of eigenvalues for $H$, and $\{\phi_r : r \in \mathbb{N}\}$ be the corresponding set of eigenfunctions for $H$. Let $\lambda(H_n) := \{\gamma_r : r = 1,\cdots,n\}$ be the set of eigenvalues for $H_n$. Let $\boldsymbol{u}_r = (u_r(X_i),\cdots,u_r(X_n))^T$ be the $r$-th eigenvector for $H_n$.

Here (and in the proofs in Appendix B-D) we assume that all eigenvalues in $\lambda(H)$ have multiplicity one for simplicity. In the case where certain eigenvalues have multiplicity larger than one and the corresponding eigenfunctions are not unique, we can always find an orthogonal matrix that transforms the set of eigenvectors $\{\boldsymbol{u}_r : r = 1,\cdots,n\}$ to match the components of $\{\phi_r : r \in \mathbb{N}\}$, as considered by Zhang et al. (2012) (Lemma 8 in [6]).

For each $r$ where $\gamma_r > 0$, define the function

$$g_{r,n}(x) := \frac{1}{\sqrt{n}\gamma_r} \sum_{j=1}^n k(x,X_j)u_r(X_j).$$

As discussed by Bengio et al. [7], the function $g_{r,n}$ can be viewed as an eigenfunction for $H_n$. In particular, it is easy to show that $g_{r,n}(X_i) = \sqrt{n}u_r(X_i)$ for each $i$.

The following proposition states the convergence of eigenvectors of the kernel matrix. The proposition and associated proof are adapted from Proposition 2 from Bengio et al. [7], while accommodating the clustered correlation among the observations.

**Proposition A.5.** *Suppose that Assumption A.1 holds for the sample $\underline{X}_n$. Assume that the conditions in Theorem A.3 hold and that, for each $r$, the function $x \mapsto g_{r,n}(x)$ converges uniformly in probability to a non-random limit function $g_{r,\infty}$ as $m \to \infty$, with $\mathbb{E}[g_{r,\infty}^2(X)] < \infty$. Then for each $r,i$, we have*

$$g_{r,n}(X_i) = \sqrt{n}u_r(X_i) \overset{p}{\to} \phi_r(X_i) \text{ as } m \to \infty.$$

*Proof.* We restrict our discussion to positive $\gamma_r$ and $\lambda_r$'s. By algebraic manipulation, we have

$$g_{r,n}(x) = \frac{1}{\sqrt{n}\gamma_r} \sum_{j=1}^{n} u_r(X_j)k(x, X_j)$$

$$= \frac{\sqrt{n}}{\gamma_r} \sum_{j=1}^{n} \Big[\frac{1}{\sqrt{n}}g_{r,n}(X_j)\Big]\frac{1}{n}k(x, X_j)$$

$$= \frac{1}{n\gamma_r} \sum_{j=1}^{n} g_{r,n}(X_j)k(x, X_j).$$

The above result shows that $g_{r,n}$ is an eigenfunction of $H_n$ with eigenvalue $\gamma_r$. Our goal is to show that $g_{r,\infty}$ is an eigenfunction of $H$ with eigenvalue $\lambda_r$. Following the argument of [7], by triangle inequality, for any fixed $x \in \mathcal{X}$, we can derive

$$\Big|g_{r,n}(x) - \frac{1}{\lambda_r}\int g_{r,\infty}(x')k(x, x')dP(x')\Big|$$

$$\leq \Big|\frac{1}{n\lambda_r}\sum_{j=1}^{n} g_{r,\infty}(X_j)k(x, X_j) - \frac{1}{\lambda_r}\int g_{r,\infty}(x')k(x, x')dP(x')\Big|$$

$$+ \Big|\frac{\lambda_r - \gamma_r}{n\lambda_r\gamma_r}\sum_{j=1}^{n} g_{r,\infty}(X_j)k(x, X_j)\Big| \tag{6}$$

$$+ \Big|\frac{1}{n\gamma_r}\sum_{j=1}^{n} k(x, X_j)[g_{r,n}(X_j) - g_{r,\infty}(X_j)]\Big|$$

$$=: A_n + B_n + C_n.$$

Next we study each of the above terms.

First, by LLN applied to clusters, we have

$$A_n = \Big|\frac{1}{n\lambda_r}\sum_{j=1}^{n} g_{r,\infty}(X_j)k(x, X_j) - \frac{1}{\lambda_r}\int g_{r,\infty}(x')k(x, x')dP(x')\Big|$$

$$= \frac{1}{d\lambda_r}\Big|\frac{1}{m}\sum_{j=1}^{m}\Big[\sum_{\ell=dj-d+1}^{dj} g_{r,\infty}(X_\ell)k(x, X_\ell)\Big] - d\int g_{r,\infty}(x')k(x, x')dP(x')\Big| \tag{7}$$

$$\xrightarrow{p} 0 \text{ as } m \to \infty.$$

Note that, since $\mathbb{E}[k^2(X', X)] < \infty$, we have $\mathbb{E}[k^2(x, X)] = \mathbb{E}[k^2(X', X)|X' = x] < \infty$ for any fixed $x \in \mathcal{X}$.

From Theorem A.3, we know that $\gamma_r \xrightarrow{p} \lambda_r$ for each $r$, and thus $\gamma_r$ is bounded in probability. Therefore, by LLN applied to clusters,

$$B_n = \Big|\frac{\lambda_r - \gamma_r}{n\lambda_r\gamma_r}\sum_{j=1}^{n} g_{r,\infty}(X_j)k(x, X_j)\Big|$$

$$\leq \Big|\frac{\lambda_r - \gamma_r}{\lambda_r\gamma_r}\Big|\Big|\frac{1}{n}\sum_{j=1}^{n} g_{r,\infty}(X_j)k(x, X_j)\Big| \tag{8}$$

$$\xrightarrow{p} 0 \times \Big|\mathbb{E}[g_{r,\infty}(X)k(x, X)]\Big| = 0 \text{ as } m \to \infty,$$

where we use Cauchy-Schwarz inequality:

$$\Big|\mathbb{E}[g_{r,\infty}(X)k(x, X)]\Big| \leq \mathbb{E}[|g_{r,\infty}(X)k(x, X)|] \leq \Big(\mathbb{E}[g_{r,\infty}^2(X)]\mathbb{E}[k^2(x, X)]\Big)^{1/2} < \infty.$$

Finally, again by LLN applied to clusters, we have

$$C_n = \left| \frac{1}{n\gamma_r} \sum_{j=1}^{n} k(x, X_j)[g_{r,n}(X_j) - g_{r,\infty}(X_j)] \right|$$

$$\leq \frac{1}{n\gamma_r} \sum_{j=1}^{n} \left| k(x, X_j)[g_{r,n}(X_j) - g_{r,\infty}(X_j)] \right|$$
(9)

$$\leq \frac{1}{\gamma_r} \sup_{x' \in \mathcal{X}} \left| g_{r,n}(x') - g_{r,\infty}(x') \right| \times \frac{1}{n} \sum_{j=1}^{n} |k(x, X_j)|$$

$$\xrightarrow{p} 0 \times \mathbb{E}[|k(x, X)|] = 0 \text{ as } m \to \infty,$$

where we use the uniform convergence assumption

$$\sup_{x' \in \mathcal{X}} \left| g_{r,n}(x') - g_{r,\infty}(x') \right| \xrightarrow{p} 0 \text{ as } m \to \infty$$

and the fact that $\mathbb{E}[|k(x, X)|] < \infty$ for any fixed $x$ (given $\mathbb{E}[k^2(x, X)] < \infty$).

By (7), (8) and (9), we see that (6) becomes

$$\left| g_{r,n}(x) - \frac{1}{\lambda_r} \int g_{r,\infty}(x') k(x, x') dP(x') \right| \xrightarrow{p} 0 \text{ as } m \to \infty,$$

i.e., $g_{r,n}(x)$ converges in probability to $\frac{1}{\lambda_r} \int g_{r,\infty}(x') k(x, x') dP(x')$ pointwise for each $x \in \mathcal{X}$.

We already know that $g_{r,n} \xrightarrow{p} g_{r,\infty}$ pointwise. By uniqueness of limit, we obtain

$$\lambda_r g_{r,\infty}(x) = \int g_{r,\infty}(x') k(x, x') dP(x'),$$

indicating that $g_{r,\infty}$ is an eigenfunction of $H$ with eigenvalue $\lambda_r$, i.e., $g_{r,\infty} = \phi_r$.

Finally, consider any observation $X_i$ from the random sample $\underline{X}_n$. By the uniform convergence assumption for $g_{r,n}(x)$, we have, for any $\epsilon > 0$,

$$\Pr\left( |g_{r,n}(X_i) - \phi_r(X_i)| > \epsilon \right) \leq \Pr\left( \sup_{x \in \mathcal{X}} |g_{r,n}(x) - \phi_r(x)| > \epsilon \right)$$

$$\to 0 \text{ as } m \to \infty.$$

Therefore, for each $r, i$, we have

$$g_{r,n}(X_i) = \sqrt{n} u_r(X_i) \xrightarrow{p} \phi_r(X_i) \text{ as } m \to \infty,$$

completing the proof. $\qquad\square$

## B  Proof of Theorem 3.2

We first introduce a lemma that will be useful in proving the theorem. The following lemma is adapted from Theorem 4.2 of [8] and Lemma 9 of [6].

**Lemma B.1.** *Let $\{A_{R,m}\}$ be a double sequence of random vectors indexed by $R$ and $m$. Let $\{B_R\}$ and $\{C_m\}$ be sequences of random vectors and $D$ be a random vector. Suppose that we have $A_{R,m} \xrightarrow{p} B_R$ as $m \to \infty$ for each $R$, and $B_R \xrightarrow{d} D$ as $R \to \infty$. Further suppose that*

$$\lim_{R \to \infty} \limsup_{m \to \infty} \Pr(\|A_{R,m} - C_m\| > \epsilon) = 0$$

*for any $\epsilon > 0$. Then $C_m \xrightarrow{d} D$ as $m \to \infty$.*

The proof of Lemma B.1 is provided in Appendix E.

## B.1 Proof of Theorem 3.2

To prove Theorem 3.2, we adopt a strategy similar to the proof for Theorem 3 of Zhang et al. (2012) [6]. Let $\boldsymbol{u}_{X,r}^* := \sqrt{\gamma_{X,r}}\boldsymbol{u}_{X,r}$ and $\boldsymbol{u}_{X,r}^* := \sqrt{\gamma_{Y,r}}\boldsymbol{u}_{Y,r}$ for each $r$. Define

$$Q_{rs} := \frac{1}{\sqrt{n}}\left(\boldsymbol{u}_{X,r}^*\right)^T\left(\boldsymbol{u}_{Y,s}^*\right) = \frac{1}{\sqrt{n}}\sum_{i=1}^n u_{X,r}^*(X_i)u_{Y,s}^*(Y_i),$$

where $u_{X,r}^*(X_i)$ and $u_{Y,r}^*(Y_i)$ are the $i$-th elements of $\boldsymbol{u}_{X,r}^*$ and $\boldsymbol{u}_{Y,r}^*$, respectively. We then note that

$$n\,\mathrm{HSIC}(P_n) = \frac{1}{n}\mathrm{tr}(\widetilde{\boldsymbol{K}}_X\widetilde{\boldsymbol{K}}_Y) = \frac{1}{n}\sum_{1\leq r,s\leq n}\left[\left(\boldsymbol{u}_{X,r}^*\right)^T\left(\boldsymbol{u}_{Y,s}^*\right)\right]^2 = \sum_{1\leq r,s\leq n}Q_{rs}^2.$$

We will break the proof of Theorem 3.2 into two parts:

**(i)** If $X \perp\!\!\!\perp Y$, then for any fixed $L \in \mathbb{N}$, we have

$$\sum_{1\leq r,s\leq L}Q_{rs}^2 \xrightarrow{d} \sum_{t=1}^{L^2}\ell_t z_t^2 \text{ as } m\to\infty, \tag{10}$$

where $z_t$'s are i.i.d. standard normal variables, and $\ell_t$'s are the eigenvalues of $\mathbb{E}[\boldsymbol{w}\boldsymbol{w}^T]$, with $\boldsymbol{w}$ being the random vector obtained by stacking the columns in the $L \times L$ matrix $\boldsymbol{N}$, whose $(r,s)$-th entry is

$$N_{rs} = \frac{1}{\sqrt{d}}\sum_{i=1}^d\sqrt{\lambda_{X,r}\lambda_{Y,s}}\phi_{X,r}(X_i)\phi_{Y,s}(Y_i).$$

**(ii)** The result (10) still holds when $L = n \to \infty$, which is satisfied as $m\to\infty$. In other words,

$$n\,\mathrm{HSIC}(P_n) = \sum_{1\leq r,s\leq n}Q_{rs}^2 \xrightarrow{d} \sum_{t=1}^{\infty}\ell_t z_t^2 \text{ as } m\to\infty,$$

where $\ell_t$'s are now the eigenvalues of the infinite matrix $\mathbb{E}[\boldsymbol{w}_\infty\boldsymbol{w}_\infty^T]$, with $\boldsymbol{w}_\infty$ being the infinite random vector whose elements are of the form:

$$\frac{1}{\sqrt{d}}\sum_{i=1}^d\sqrt{\lambda_{X,r}\lambda_{Y,s}}\phi_{X,r}(X_i)\phi_{Y,s}(Y_i) \quad \text{for } r,s \in \mathbb{N}.$$

Alternatively, $\ell_t$'s can be viewed as the solutions to the eigenvalue problem

$$\ell_t\psi_{t,rs}$$
$$= \frac{1}{d}\sum_{p,q=1}^{\infty}\mathbb{E}\left[\left(\sum_{i=1}^d\sqrt{\lambda_{X,r}\lambda_{Y,s}}\phi_{X,r}(X_i)\phi_{Y,s}(Y_i)\right)\left(\sum_{i=1}^d\sqrt{\lambda_{X,p}\lambda_{Y,q}}\phi_{X,p}(X_i)\phi_{Y,q}(Y_i)\right)\right]\psi_{t,pq}$$

for some double sequence $\{\psi_{t,rs}\}_{r,s=1}^{\infty} \in \mathbb{R}$.

We focus on proving part **(i)** of the theorem. Assume that the null hypothesis $H_0 : X \perp\!\!\!\perp Y$ hold, and consider a fixed $L \in \mathbb{N}$.

For $i = 1,\cdots,m$, let $\boldsymbol{v}_i$ be the random vector obtained by stacking the columns in the $L \times L$ matrix $\boldsymbol{M}_i$, whose $(r,s)$-th entry is

$$M_{i,rs} = \frac{1}{\sqrt{d}}\sum_{j=di-d+1}^{di}u_{X,r}^*(X_j)u_{Y,s}^*(Y_j).$$

Let $\boldsymbol{w}_i$ be the random vector obtained by stacking the columns in the $L \times L$ matrix $\boldsymbol{N}_i$, whose $(r,s)$-th entry is

$$N_{i,rs} = \frac{1}{\sqrt{d}}\sum_{j=di-d+1}^{di}\sqrt{\lambda_{X,r}\lambda_{Y,s}}\phi_{X,r}(X_j)\phi_{Y,s}(Y_j).$$

Since $X \perp\!\!\!\perp Y$, we have

$$\mathbb{E}[\phi_{X,r}(X_i)\phi_{Y,s}(Y_i)] = \mathbb{E}[\phi_{X,r}(X_i)]\,\mathbb{E}[\phi_{Y,s}(Y_i)] = 0 \text{ for all } i,$$

where we use the assumption that the kernels $\tilde{k}_X$ and $\tilde{k}_Y$ are centered. Therefore, we have $\mathbb{E}[\boldsymbol{w}] = \mathbf{0}$ and it follows that $\mathbb{C}\text{ov}[\boldsymbol{w}] = \mathbb{E}[\boldsymbol{w}\boldsymbol{w}^T]$.

By multivariate central limit theorem, we then have

$$\frac{1}{\sqrt{m}}\sum_{i=1}^{m} \boldsymbol{w}_i \xrightarrow{d} N(\mathbf{0}, \mathbb{E}[\boldsymbol{w}\boldsymbol{w}^T]) \text{ as } m \to \infty.$$

By Theorem A.3 and Proposition A.5, for any $X_i, Y_i$ and any fixed $r$, we have [1]

$$\frac{1}{n}\gamma_{X,r} \xrightarrow{P} \lambda_{X,r}, \quad \frac{1}{n}\gamma_{Y,r} \xrightarrow{P} \lambda_{Y,r} \text{ as } m \to \infty;$$

$$\sqrt{n}u_{X,r}(X_i) \xrightarrow{P} \phi_{X,r}(X_i), \quad \sqrt{n}u_{Y,r}(Y_i) \xrightarrow{P} \phi_{Y,r}(Y_i) \text{ as } m \to \infty.$$

Recall that $u_{X,r}^*(X_i) = \sqrt{\gamma_{X,r}}u_{X,r}(X_i)$ and $u_{Y,r}^*(Y_i) = \sqrt{\gamma_{Y,r}}u_{Y,r}(Y_i)$. Therefore, by continuous mapping theorem and Slutsky's theorem, for any fixed $r$ and $s$,

$$u_{X,r}^*(X_i)u_{Y,s}^*(Y_i) \xrightarrow{P} \sqrt{\lambda_{X,r}\lambda_{Y,s}}\phi_{X,r}(X_i)\phi_{Y,s}(Y_i) \text{ as } m \to \infty.$$

As a consequence, we have $\boldsymbol{v}_i \xrightarrow{P} \boldsymbol{w}_i$ for each $i$. Using Lemma B.1 with $A_{R,m} = \frac{1}{\sqrt{R}}\sum_{i=1}^{R}\boldsymbol{v}_i$, $B_R = \frac{1}{\sqrt{R}}\sum_{i=1}^{R}\boldsymbol{w}_i$, $C_m = \frac{1}{\sqrt{m}}\sum_{i=1}^{m}\boldsymbol{v}_i$ and $D \sim \mathcal{N}(\mathbf{0}, \mathbb{E}[\boldsymbol{w}\boldsymbol{w}^T])$, we can derive

$$\frac{1}{\sqrt{m}}\sum_{i=1}^{m} \boldsymbol{v}_i \xrightarrow{d} \mathcal{N}(\mathbf{0}, \mathbb{E}[\boldsymbol{w}\boldsymbol{w}^T]) \text{ as } m \to \infty.$$

We perform an eigendecomposition of $\mathbb{E}[\boldsymbol{w}\boldsymbol{w}^T]$ such that $\mathbb{E}[\boldsymbol{w}\boldsymbol{w}^T] = \boldsymbol{U}\boldsymbol{\Lambda}\boldsymbol{U}^T$, where $\boldsymbol{\Lambda}$ is a diagonal matrix containing the eigenvalues of $\mathbb{E}[\boldsymbol{w}\boldsymbol{w}^T]$ and $\boldsymbol{U}$ is an orthogonal matrix.

Let $\boldsymbol{Z} = \boldsymbol{U}^T\left[\frac{1}{\sqrt{m}}\sum_{i=1}^{m}\boldsymbol{v}_i\right]$. Then by continuous mapping theorem, we have

$$\boldsymbol{Z} \xrightarrow{d} \mathcal{N}(\mathbf{0}, \boldsymbol{U}^T\mathbb{E}[\boldsymbol{w}\boldsymbol{w}^T]\boldsymbol{U}) = \mathcal{N}(\mathbf{0}, \boldsymbol{U}^T\boldsymbol{U}\boldsymbol{\Lambda}\boldsymbol{U}^T\boldsymbol{U}) = \mathcal{N}(\mathbf{0}, \boldsymbol{\Lambda}) \text{ as } m \to \infty.$$

It follows that

$$\sum_{1\le r,s\le L} Q_{rs}^2 = \sum_{1\le r,s\le L}\left[\frac{1}{\sqrt{n}}\sum_{i=1}^{n} u_{X,r}^*(X_i)u_{Y,s}^*(Y_i)\right]^2$$

$$= \sum_{1\le r,s\le L}\left[\frac{1}{\sqrt{m}}\sum_{i=1}^{m}\frac{1}{\sqrt{d}}\sum_{j=di-d+1}^{di} u_{X,r}^*(X_j)u_{Y,s}^*(Y_j)\right]^2$$

$$= \left[\frac{1}{\sqrt{m}}\sum_{i=1}^{m}\boldsymbol{v}_i\right]^T\left[\frac{1}{\sqrt{m}}\sum_{i=1}^{m}\boldsymbol{v}_i\right]$$

$$= \left[\frac{1}{\sqrt{m}}\sum_{i=1}^{m}\boldsymbol{v}_i\right]^T\boldsymbol{U}\boldsymbol{U}^T\left[\frac{1}{\sqrt{m}}\sum_{i=1}^{m}\boldsymbol{v}_i\right]$$

$$= \boldsymbol{Z}^T\boldsymbol{Z} \xrightarrow{d} \sum_{t=1}^{L^2} \ell_t z_t^2 \text{ as } m \to \infty,$$

where $\ell_t$'s are the eigenvalues of $\mathbb{E}[\boldsymbol{w}\boldsymbol{w}^T]$ and $z_t$'s are i.i.d. standard normal variables.

To prove part **(ii)** of the theorem, we can use Lemma 9 from Zhang et al. (2012) [6]. The argument would be similar to that in [6] and we skip the details here.

---

[1]Here we assume that the $(i,j)$-th entry of the centered kernel matrix $\widetilde{\boldsymbol{K}}_X$ ($\widetilde{\boldsymbol{K}}_Y$) well approximates $\tilde{k}_X(X_i, X_j)$ ($\tilde{k}_Y(Y_i, Y_j)$). To address data dependence of the entries of $\widetilde{\boldsymbol{K}}_X$ and $\widetilde{\boldsymbol{K}}_Y$, we could show that, under regularity conditions, these empirically centered kernel functions converge uniformly in probability to $\tilde{k}_X$ and $\tilde{k}_Y$, as considered in Proposition 2 of [7].

## C Proof of Theorem 3.3

We first introduce a lemma that will help with the proof. The following lemma is in the same spirit as Lemma B.1, while replacing convergence in distribution with convergence in probability in assumptions and results.

**Lemma C.1.** *Let $\{A_{R,m}\}$ be a double sequence of random vectors indexed by $R$ and $m$. Let $\{B_R\}$ and $\{C_m\}$ be sequences of random vectors and $D$ be a random vector. Suppose that we have $A_{R,m} \xrightarrow{p} B_R$ as $m \to \infty$ for each $R$, and $B_R \xrightarrow{p} D$ as $R \to \infty$. Further suppose that*

$$\lim_{R\to\infty} \limsup_{m\to\infty} \Pr(\|A_{R,m} - C_m\| > \epsilon) = 0$$

*for any $\epsilon > 0$. Then $C_m \xrightarrow{p} D$ as $m \to \infty$.*

The proof of Lemma C.1 is provided in Appendix E.

### C.1 Proof of Theorem 3.3

To begin the proof for Theorem 3.3, we assume that there exists some $r, s \in \mathbb{N}$ such that $\mathbb{E}[\phi_{X,r}(X)\phi_{Y,s}(Y)] \neq 0$. We can find a fixed $L \in \mathbb{N}$ such that $r, s \leq L$. Let $Q_{rs}$, $\boldsymbol{w}, \boldsymbol{w}_i, \boldsymbol{v}_i$ be defined as in the proof of Theorem 3.2 (Appendix B.1).

Since $\mathbb{E}[\phi_{X,r}(X)\phi_{Y,s}(Y)] \neq 0$, we have

$$\mathbb{E}\left[\frac{1}{\sqrt{d}} \sum_{i=1}^{d} \sqrt{\lambda_{X,r}\lambda_{Y,s}}\phi_{X,r}(X_i)\phi_{Y,s}(Y_i)\right] = \sqrt{d}\,\mathbb{E}\left[\sqrt{\lambda_{X,r}\lambda_{Y,s}}\phi_{X,r}(X)\phi_{Y,s}(Y)\right] \neq 0.$$

It follows that $\mathbb{E}[\boldsymbol{w}] \neq 0$. By weak law of large numbers, we have

$$\frac{1}{m} \sum_{i=1}^{m} \boldsymbol{w}_i \xrightarrow{p} \mathbb{E}[\boldsymbol{w}] \text{ as } m \to \infty.$$

Using Theorem A.3, Proposition A.5 and Lemma C.1 with $A_{R,m} = \frac{1}{R}\sum_{i=1}^{R}\boldsymbol{v}_i$, $B_R = \frac{1}{R}\sum_{i=1}^{R}\boldsymbol{w}_i$, $C_m = \frac{1}{m}\sum_{i=1}^{m}\boldsymbol{v}_i$ and $D = \mathbb{E}[\boldsymbol{w}]$, we can derive

$$\frac{1}{m} \sum_{i=1}^{m} \boldsymbol{v}_i \xrightarrow{p} \mathbb{E}[\boldsymbol{w}] \text{ as } m \to \infty.$$

Therefore, by continuous mapping theorem,

$$\frac{1}{m} \sum_{1 \leq r,s \leq L} Q_{rs}^2 = \left[\frac{1}{m}\sum_{i=1}^{m}\boldsymbol{v}_i\right]^T \left[\frac{1}{m}\sum_{i=1}^{m}\boldsymbol{v}_i\right]$$

$$\xrightarrow{p} \mathbb{E}[\boldsymbol{w}]^T \mathbb{E}[\boldsymbol{w}] > 0 \text{ as } m \to \infty.$$

As a consequence,

$$\sum_{1 \leq r,s \leq L} Q_{rs}^2 \xrightarrow{p} \infty \text{ as } m \to \infty.$$

For a fixed $L$, we can always find a large enough $m$ such that $n = md \geq L$ and thus

$$\frac{1}{n} \text{tr}(\widetilde{\boldsymbol{K}}_X \widetilde{\boldsymbol{K}}_Y) = \sum_{1 \leq r,s \leq n} Q_{rs}^2 \geq \sum_{1 \leq r,s \leq L} Q_{rs}^2.$$

It follows that

$$n \, \text{HSIC}(P_n) = \frac{1}{n} \text{tr}(\widetilde{\boldsymbol{K}}_X \widetilde{\boldsymbol{K}}_Y) \xrightarrow{p} \infty \text{ as } m \to \infty.$$

Finally, we show that, when $\tilde{k}_X$ and $\tilde{k}_Y$ are characteristic kernels [9],

$$\mathbb{E}[\phi_{X,r}(X)\phi_{Y,s}(Y)] \neq 0 \text{ for some } r, s \in \mathbb{N} \iff X \not\perp\!\!\!\perp Y.$$

(1) Given $X \perp\!\!\!\perp Y$, then $\mathbb{E}[\phi_{X,r}(X)\phi_{Y,s}(Y)] = \mathbb{E}[\phi_{X,r}(X)]\,\mathbb{E}[\phi_{Y,s}(Y)] = 0$ for all $r, s \in \mathbb{N}$. As a contrapositive, $\mathbb{E}[\phi_{X,r}(X)\phi_{Y,s}(Y)] \neq 0$ for some $r, s \in \mathbb{N}$ implies $X \not\perp\!\!\!\perp Y$. This holds true regardless of the kernels being used.

(2) Given that $\mathbb{E}[\phi_{X,r}(X)\phi_{Y,s}(Y)] = 0$ for all $r, s \in \mathbb{N}$. Then for any $f \in \widetilde{\mathcal{H}}_X$, $g \in \widetilde{\mathcal{H}}_Y$, we have $\mathbb{Cov}(f(X), g(Y)) = \mathbb{E}[f(X)g(Y)] = 0$, since any $f$ and $g$ can be expressed as linear combinations of $\phi_{X,r}$'s and $\phi_{Y,s}$'s, respectively. Hence, the largest singular value of $C_{XY}$, which is the maximized covariance between functions in $\widetilde{\mathcal{H}}_X$ and $\widetilde{\mathcal{H}}_Y$, must be zero. Consequently, the squared Hilbert-Schmidt norm of $C_{XY}$, $\|C_{XY}\|_{HS}^2 \equiv \mathrm{HSIC}(P_{XY})$, is also zero. When $\tilde{k}_X$ and $\tilde{k}_Y$ are characteristic kernels, $\mathrm{HSIC}(P_{XY}) = 0$ if and only if $X \perp\!\!\!\perp Y$ [9]. As a result, $\mathrm{HSIC}(P_{XY}) = 0$ implies that $X \perp\!\!\!\perp Y$.

As a contrapositive, $X \not\perp\!\!\!\perp Y$ implies $\mathbb{E}[\phi_{X,r}(X)\phi_{Y,s}(Y)] \neq 0$ for some $r, s \in \mathbb{N}$.

## D Proof of Proposition 3.4

Assume that the conditions in Theorem 3.2 hold. We would like to show that, under the null hypothesis $H_0 : X \perp\!\!\!\perp Y$, the statistic $n\,\mathrm{HSIC}(P_n) = \frac{1}{n}\mathrm{tr}(\widetilde{K}_X\widetilde{K}_Y)$ has the same asymptotic distribution as

$$\widetilde{T} = \frac{1}{m}\sum_{t=1}^{n^2}\tilde{\ell}_t z_t^2,$$

where $z_t$'s are i.i.d. standard normal variables and $\tilde{\ell}_t$'s are eigenvalues of $\widetilde{V}\widetilde{V}^T$, with $\widetilde{V} = [\tilde{v}_1, \cdots, \tilde{v}_m]$. Each vector $\tilde{v}_i$ is obtained by stacking the columns in the $n \times n$ matrix $\widetilde{M}_i$, whose $(r, s)$-th entry is

$$\widetilde{M}_{i,rs} = \frac{1}{\sqrt{d}}\sum_{j=di-d+1}^{di} u_{X,r}^*(X_j)u_{Y,s}^*(Y_j),$$

where $u_{X,r}^*(X_j)$ is the $j$-th element of $\boldsymbol{u}_{X,r}^* = \sqrt{\gamma_{X,r}}\boldsymbol{u}_{X,r}$, and $u_{Y,s}^*(Y_j)$ is the $j$-th element of $\boldsymbol{u}_{Y,s}^* = \sqrt{\gamma_{Y,s}}\boldsymbol{u}_{Y,s}$.

Here we present a sketch proof extended from the proof for Theorem 1 of Gretton et al. (2009) [10]. From Theorem 3.2, we have that

$$n\,\mathrm{HSIC}(P_n) = \frac{1}{n}\mathrm{tr}\left(\widetilde{K}_X\widetilde{K}_Y\right) \xrightarrow{d} \sum_{t=1}^{\infty}\ell_t z_t^2 \text{ as } m \to \infty,$$

where $\ell_t$'s are the eigenvalues of the infinite matrix $\boldsymbol{\Sigma} := \mathbb{E}[\boldsymbol{w}_\infty \boldsymbol{w}_\infty^T]$, with $\boldsymbol{w}_\infty$ being the infinite random vector whose elements are of the form:

$$\frac{1}{\sqrt{d}}\sum_{i=1}^{d}\sqrt{\lambda_{X,r}\lambda_{Y,s}}\phi_{X,r}(X_i)\phi_{Y,s}(Y_i) \quad \text{for } r, s \in \mathbb{N}.$$

A natural estimator for $\ell_t$'s is the set of eigenvalues for the empirical matrix $\hat{\boldsymbol{\Sigma}}$, given by

$$\hat{\boldsymbol{\Sigma}} := \frac{1}{m}\sum_{i=1}^{m}\tilde{v}_i\tilde{v}_i^T$$

$$= \frac{1}{m}\begin{pmatrix}\tilde{v}_1 & \cdots & \tilde{v}_m\end{pmatrix}\begin{pmatrix}\tilde{v}_1^T \\ \vdots \\ \tilde{v}_m^T\end{pmatrix}$$

$$= \frac{1}{m}\widetilde{V}\widetilde{V}^T.$$

Letting $\tilde{\ell}_t$'s be the eigenvalues of $\widetilde{V}\widetilde{V}^T$, we would like to show that

$$\sum_{t=1}^{\infty}\left(\frac{1}{m}\tilde{\ell}_t - \ell_t\right)z_t^2 \xrightarrow{p} 0 \text{ as } m \to \infty. \tag{11}$$

Following the proof of Theorem 1 in [10], the key step to establish (11) is to show that $\sum_t |\frac{1}{m}\tilde{\ell}_t - \ell_t| \xrightarrow{p} 0$ as $m \to \infty$.

By an extension of the Hoffman–Wielandt inequality, we have

$$\sum_{t=1}^{\infty} \left| \frac{1}{m}\tilde{\ell}_t - \ell_t \right| \leq \|\hat{\boldsymbol{\Sigma}} - \boldsymbol{\Sigma}\|_1,$$

where $\| \cdot \|_1$ is the trace norm (the sum of singular values of the operator).

For $i = 1, \cdots, m$, let $\tilde{\boldsymbol{w}}_i$ be the random vector obtained by stacking the columns in the $n \times n$ matrix $\widetilde{\boldsymbol{N}}_i$, whose $(r, s)$-th entry is

$$\widetilde{N}_{i,rs} = \frac{1}{\sqrt{d}} \sum_{j=di-d+1}^{di} \sqrt{\lambda_{X,r}\lambda_{Y,s}}\phi_{X,r}(X_j)\phi_{Y,s}(Y_j).$$

We can then write

$$
\begin{aligned}
\|\hat{\boldsymbol{\Sigma}} - \boldsymbol{\Sigma}\|_1 &= \left\| \frac{1}{m}\sum_{i=1}^{m} \tilde{\boldsymbol{v}}_i\tilde{\boldsymbol{v}}_i^T - \mathbb{E}[\boldsymbol{w}_\infty \boldsymbol{w}_\infty^T] \right\|_1 \\
&\leq \left\| \frac{1}{m}\sum_{i=1}^{m} \tilde{\boldsymbol{v}}_i\tilde{\boldsymbol{v}}_i^T - \frac{1}{m}\sum_{i=1}^{m} \tilde{\boldsymbol{w}}_i\tilde{\boldsymbol{w}}_i^T \right\|_1 + \left\| \frac{1}{m}\sum_{i=1}^{m} \tilde{\boldsymbol{w}}_i\tilde{\boldsymbol{w}}_i^T - \mathbb{E}[\tilde{\boldsymbol{w}}_1\tilde{\boldsymbol{w}}_1^T] \right\|_1 \\
&\quad + \left\| \mathbb{E}[\tilde{\boldsymbol{w}}_1\tilde{\boldsymbol{w}}_1^T] - \mathbb{E}[\boldsymbol{w}_\infty \boldsymbol{w}_\infty^T] \right\|_1 \\
&=: A_n + B_n + C_n.
\end{aligned}
$$

We can show that $A_n \xrightarrow{p} 0$ as $m \to \infty$ due to the convergence of the eigenvalues and eigenvector elements of $\widetilde{\boldsymbol{K}}_X$ and $\widetilde{\boldsymbol{K}}_Y$ to the eigenvalues and eigenfunctions of $\tilde{k}_X$ and $\tilde{k}_Y$, using Theorem A.3 and Proposition A.5. Furthermore, $B_n \xrightarrow{p} 0$ as $m \to \infty$ due to Proposition 12 from [11]. Finally, $C_n \to 0$ as $m \to \infty$ due to the convergence of the finite truncation of a linear operator (e.g., see Proposition 2.1 from [12]). As a consequence, we have

$$\sum_{t=1}^{\infty} \left| \frac{1}{m}\tilde{\ell}_t - \ell_t \right| \leq \|\hat{\boldsymbol{\Sigma}} - \boldsymbol{\Sigma}\|_1 \xrightarrow{p} 0 \text{ as } m \to \infty,$$

which gives us

$$\widetilde{T} = \frac{1}{m}\sum_{t=1}^{n^2} \tilde{\ell}_t z_t^2 \xrightarrow{p} \sum_{t=1}^{\infty} \ell_t z_t^2 \text{ as } m \to \infty,$$

completing the proof.

# E   Proofs of the lemmas

**Lemma A.2.** *Suppose that Assumption A.1 holds for the sample $\underline{X}_n$. Let $V_n$ be a V-statistic based on $\underline{X}_n$ with a bivariate symmetric kernel function $f(x, x')$:*

$$V_n := \frac{1}{n^2}\sum_{i=1}^{n}\sum_{j=1}^{n} f(X_i, X_j).$$

*Assume that $V_n$ is non-degenerate and the class $\mathcal{C} := \{x \mapsto f(x, x') : x' \in \mathcal{X}\}$ is a P-Donsker class, then*

$$V_n \xrightarrow{p} \mathbb{E}[f(X, X')] \text{ as } m \to \infty,$$

*where $X'$ is an independent copy of $X$.*

*Proof.* We utilize empirical process theory in this proof. For a given bivariate function $f$, we use the notation $Pf$ to denote the function

$$x \mapsto \int f(x, x')dP(x'),$$

and we define $P_1 P_2 f$ for any set of probability measures $P_1, P_2$ as the mapping

$$(x, x') \mapsto \int \int f(x, x') dP_2(x') dP_1(x).$$

Using these notations, we can write $V_n = P_n^2 f$ and $\mathbb{E}[f(X, X')] = P^2 f$.

By symmetry of $f$, we have

$$\begin{aligned}
V_n = P_n^2 f &= P^2 f + P_n^2 f - P^2 f \\
&= P^2 f + P_n(P_n - P)f + (P_n - P)Pf \\
&= P^2 f + (P_n - P)(P_n - P)f + P(P_n - P)f + (P_n - P)Pf \\
&= P^2 f + 2(P_n - P)Pf + (P_n - P)^2 f.
\end{aligned} \tag{12}$$

Letting $f_1 := Pf$, note that

$$\begin{aligned}
(P_n - P)Pf &= \frac{1}{n} \sum_{i=1}^{n} \Big( f_1(X_i) - \mathbb{E}[f_1(X)] \Big) \\
&= \frac{1}{md} \sum_{i=1}^{m} \sum_{j=di-d+1}^{di} \Big( f_1(X_j) - \mathbb{E}[f_1(X)] \Big) \\
&= \frac{1}{m} \sum_{i=1}^{m} \Big( \frac{1}{d} \sum_{j=di-d+1}^{di} f_1(X_j) - \mathbb{E}[f_1(X)] \Big) \\
&\xrightarrow{p} \mathbb{E}\Big[ \frac{1}{d} \sum_{j=1}^{d} f_1(X_j) - \mathbb{E}[f_1(X)] \Big] = \mathbb{E}[f_1(X)] - \mathbb{E}[f_1(X)] = 0 \text{ as } m \to \infty.
\end{aligned} \tag{13}$$

We now show that $(P_n - P)^2 f$ is also asymptotically negligible.

Letting $f_{1n} := (P_n - P)f$, we have

$$\begin{aligned}
\sup_{x \in \mathcal{X}} |f_{1n}(x)| = \sup_{x \in \mathcal{X}} \Big| \int f(x, x') d(P_n - P)(x') \Big| &= \sup_{x \in \mathcal{X}} \Big| \int f(x', x) d(P_n - P)(x') \Big| \\
&\leq \frac{1}{d} \sup_{x \in \mathcal{X}} \Big| \frac{1}{m} \sum_{i=1}^{m} f(X_{di-d+1}, x) - Pf \Big| + \cdots + \frac{1}{d} \sup_{x \in \mathcal{X}} \Big| \frac{1}{m} \sum_{i=1}^{m} f(X_{di}, x) - Pf \Big|.
\end{aligned} \tag{14}$$

Since $\mathcal{C} = \{x \mapsto f(x, x') : x' \in \mathcal{X}\}$ is a $P$-Donsker class, it is also a $P$-Glivenko-Cantelli class (see Chapter 19.2 of [2] for definition of these classes). Therefore, by definition of a Glivenko-Cantelli class, each element in (14) would converge to 0 in probability as $m \to \infty$. As a result,

$$\sup_{x \in \mathcal{X}} |f_{1n}(x)| \xrightarrow{p} 0 \text{ as } m \to \infty.$$

Since $Pf_{1n}^2 \leq \Big[ \sup_{x \in \mathcal{X}} |f_{1n}(x)| \Big]^2$, it follows that

$$Pf_{1n}^2 \xrightarrow{p} 0 \text{ as } m \to \infty. \tag{15}$$

Next, note that $x \mapsto \int f(x, x') dP_n(x')$ is in the closure of the convex hull of $\mathcal{C}$, and $x \mapsto \int f(x, x') dP(x')$ is a fixed function. By Theorems 2.10.2 and 2.10.3 of [13], $f_{1n}(x) = \int f(x, x') d(P_n - P)(x')$ also falls in a $P$-Donsker class.

Finally, combining the above result ($f_{1n}$ belongs to a $P$-Donsker class) with (15), by Lemma 19.24 of [2], we have

$$\begin{aligned}
(P_n - P)^2 f = (P_n - P)f_{1n} &= \frac{1}{d} \Big[ \frac{1}{m} \sum_{i=1}^{m} f_{1n}(X_{di-d+1}) - Pf_{1n} \Big] + \cdots + \frac{1}{d} \Big[ \frac{1}{m} \sum_{i=1}^{m} f_{1n}(X_{di}) - Pf_{1n} \Big] \\
&= o_P(m^{-1/2}).
\end{aligned} \tag{16}$$

By (12), (13) and (16), we have that

$$V_n \xrightarrow{p} \mathbb{E}[f(X, X')] \text{ as } m \to \infty,$$

thus completing the proof. $\qquad\square$

**Lemma B.1.** *Let $\{A_{R,m}\}$ be a double sequence of random vectors indexed by $R$ and $m$. Let $\{B_R\}$ and $\{C_m\}$ be sequences of random vectors and $D$ be a random vector. Suppose that we have $A_{R,m} \xrightarrow{p} B_R$ as $m \to \infty$ for each $R$, and $B_R \xrightarrow{d} D$ as $R \to \infty$. Further suppose that*

$$\lim_{R\to\infty} \limsup_{m\to\infty} \Pr(\|A_{R,m} - C_m\| > \epsilon) = 0$$

*for any $\epsilon > 0$. Then $C_m \xrightarrow{d} D$ as $m \to \infty$.*

*Proof.* Since $A_{R,m} \xrightarrow{p} B_R$ as $m \to \infty$ for each R, we have $A_{R,m} \xrightarrow{d} B_R$ as $m \to \infty$ for each R. The rest follows from the proof for Theorem 4.2 of [8]. $\qquad\square$

**Lemma C.1.** *Let $\{A_{R,m}\}$ be a double sequence of random vectors indexed by $R$ and $m$. Let $\{B_R\}$ and $\{C_m\}$ be sequences of random vectors and $D$ be a random vector. Suppose that we have $A_{R,m} \xrightarrow{p} B_R$ as $m \to \infty$ for each $R$, and $B_R \xrightarrow{p} D$ as $R \to \infty$. Further suppose that*

$$\lim_{R\to\infty} \limsup_{m\to\infty} \Pr(\|A_{R,m} - C_m\| > \epsilon) = 0$$

*for any $\epsilon > 0$. Then $C_m \xrightarrow{p} D$ as $m \to \infty$.*

*Proof.* Given $\epsilon > 0$, we have that

$$\Pr(\|C_m - D\| > \epsilon) \leq \Pr(\|C_m - A_{R,m}\| > \epsilon/3) + \Pr(\|A_{R,m} - B_R\| > \epsilon/3) \\ + \Pr(\|B_R - D\| > \epsilon/3).$$

Fixing $R$ and letting $m \to \infty$, we have

$$\lim_{m\to\infty} \Pr(\|C_m - D\| > \epsilon)$$
$$\leq \limsup_{m\to\infty} \Pr(\|C_m - A_{R,m}\| > \epsilon/3) + \lim_{m\to\infty} \Pr(\|A_{R,m} - B_R\| > \epsilon/3) + \Pr(\|B_R - D\| > \epsilon/3)$$
$$= \limsup_{m\to\infty} \Pr(\|C_m - A_{R,m}\| > \epsilon/3) + \Pr(\|B_R - D\| > \epsilon/3),$$

where we use the fact that $A_{R,m} \xrightarrow{p} B_R$ as $m \to \infty$ for fixed $R$. Now letting $R \to \infty$, we have

$$\lim_{m\to\infty} \Pr(\|C_m - D\| > \epsilon)$$
$$\leq \lim_{R\to\infty} \limsup_{m\to\infty} \Pr(\|C_m - A_{R,m}\| > \epsilon/3) + \lim_{R\to\infty} \Pr(\|B_R - D\| > \epsilon/3)$$
$$= 0,$$

where we use $B_R \xrightarrow{p} D$ as $R \to \infty$ and $\lim_{R\to\infty} \limsup_{m\to\infty} \Pr(\|A_{R,m} - C_m\| > \epsilon) = 0$. Therefore, we have shown that $C_m \xrightarrow{p} D$ as $m \to \infty$.

$\qquad\square$

## F    Additional simulations

### F.1    Type I error simulation in non-normal data

To examine the type I error control of **HSIC$_{\text{cl}}$** in the presence of non-normal data, we modify Model (3) in the main text such that the variable $Y$ has a non-normal distribution under the null hypothesis.

Following the general simulation setting in Section 4.1.1, we let

$$\boldsymbol{y}_0 = (\beta_1 f(x_{r1}), \beta_1 f(x_{r2}), \beta_1 f(x_{r3}), \cdots, \beta_q f(x_{r1}), \beta_q f(x_{r2}), \beta_q f(x_{r3}))^T + \boldsymbol{\epsilon}.$$

We then generate $\boldsymbol{y} := (y_{11}, y_{12}, y_{13}, \cdots, y_{q1}, y_{q2}, y_{q3})^T$ by nonlinear transformations of $\boldsymbol{y}_0$. We consider three types of transformation functions: **Scenario A**: $f_A(y) = I\{y > 0\}$, **Scenario B**: $f_B(y) = \exp(y)$ and **Scenario C**: $f_C(y) = \sin(y)$. In each scenario, $\boldsymbol{y}$ is generated by applying the transformation function to each element of $\boldsymbol{y}_0$ such that the within-cluster correlation can be preserved to some extent. We fix $m = 500$ and $\rho_c = 0.5$.

Figure F1 shows the p-value QQ-plots for **HSIC$_{\text{cl}}$** and **HSIC$_{\text{orig}}$** under Type I error simulation for all three scenarios.

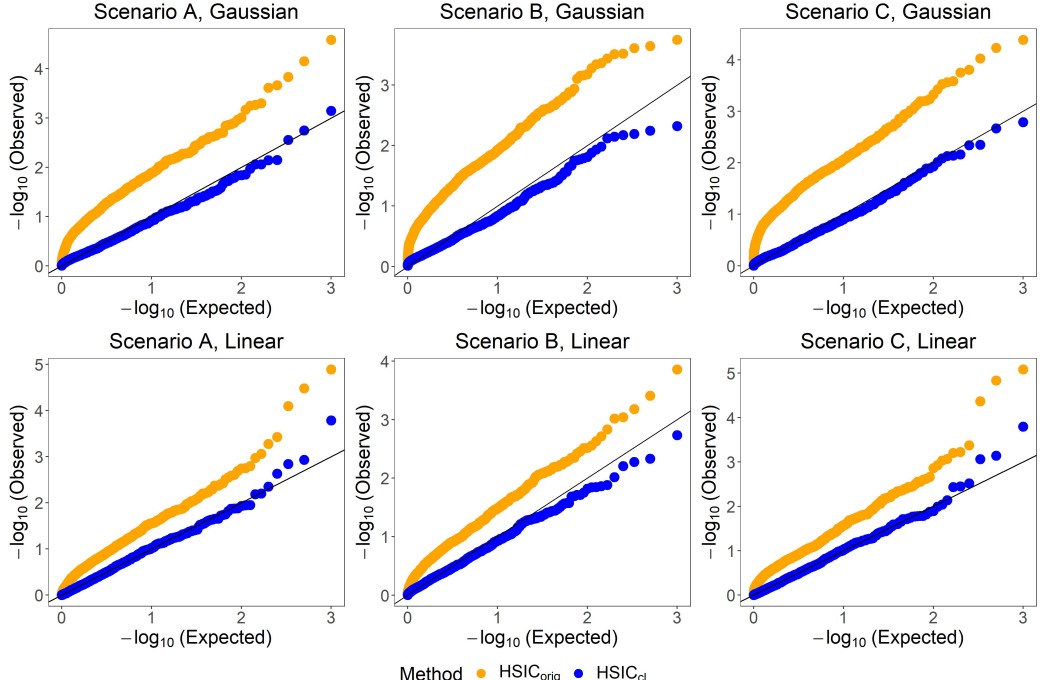

Figure F1: P-value QQ-plots for **HSIC$_{\text{cl}}$** and **HSIC$_{\text{orig}}$** under Type I error simulation for three non-normal data scenarios. Simulation parameters are set as: $m = 500$, $d = 3$ and $\rho_c = 0.5$. The top row shows results based on the Gaussian kernel, and the bottom row shows results based on the linear kernel.

## F.2 Effect of cluster size on performance of HSIC$_{\text{cl}}$

We investigate the effect of cluster size on the performance of **HSIC$_{\text{cl}}$**. We use the same simulation setting as in Section 4.1.1, where we fix $m = 500$ and $\rho_c = 0.5$ and consider different cluster sizes: $d = 2, 3, 4$ or $5$. We evaluate both type I error control and power of **HSIC$_{\text{cl}}$**. In the power simulation, we consider **Power Scenario 1** and set $\eta = 20\%$.

Figure F2 shows the p-value QQ-plots for **HSIC$_{\text{cl}}$** under Type I error simulation with different cluster sizes, when the Gaussian kernel is used. Figure F3 shows the p-value QQ-plots for **HSIC$_{\text{cl}}$** under Type I error simulation with different cluster sizes, when the linear kernel is used.

Figure F4 shows the empirical power of **HSIC$_{\text{cl}}$** under different cluster sizes, for both the Gaussian kernel and the linear kernel.

## F.3 Comparison of HSIC$_{\text{cl}}$ against HSIC$_{\text{perm}}$

An alternative way to assess the significance of the HSIC statistic is to compare the observed statistic against its permutation distribution, which could approximate the sampling distribution of the test statistic under the null hypothesis. This approach does not rely on asymptotic results and is suitable for small sample sizes. We implement a permutation-based HSIC test for clustered data, **HSIC$_{\text{perm}}$**. We construct the empirical permutation distribution of HSIC in the following way: in each permutation, we randomly shuffle the clusters for one variable and then re-construct the HSIC statistic using the

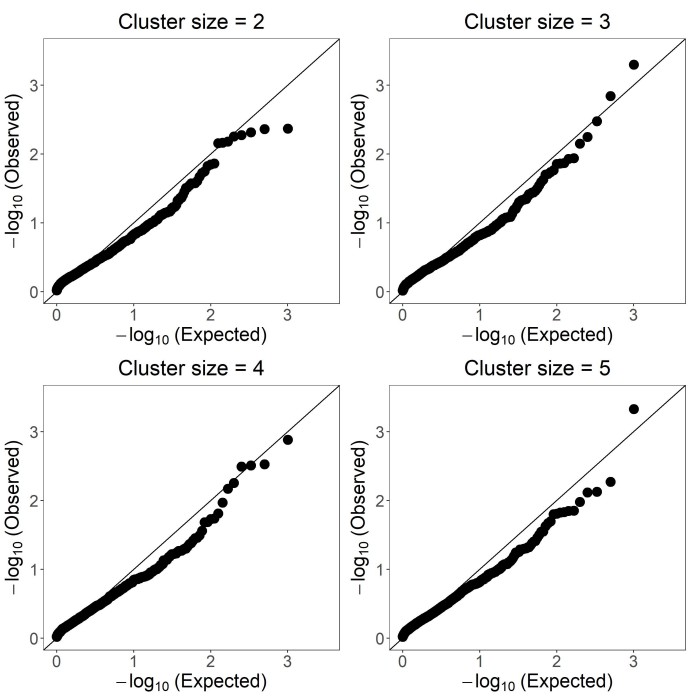

Figure F2: P-value QQ-plots for $\mathbf{HSIC_{cl}}$ under Type I error simulation with different cluster sizes. The Gaussian kernel is used. Simulation parameters are set as: $m = 500$, $\rho_c = 0.5$.

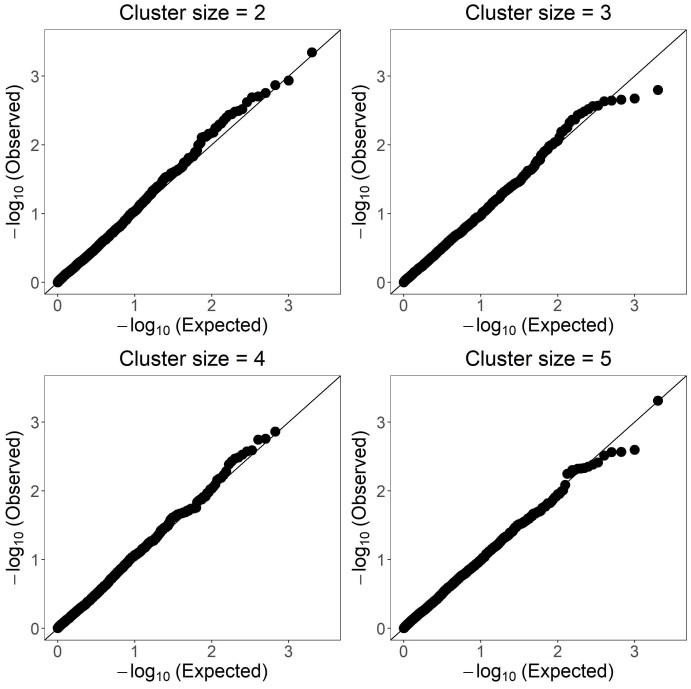

Figure F3: P-value QQ-plots for $\mathbf{HSIC_{cl}}$ under Type I error simulation with different cluster sizes. The linear kernel is used. Simulation parameters are set as: $m = 500$, $\rho_c = 0.5$.

shuffled observations; this procedure is repeated many times to obtain an empirical permutation

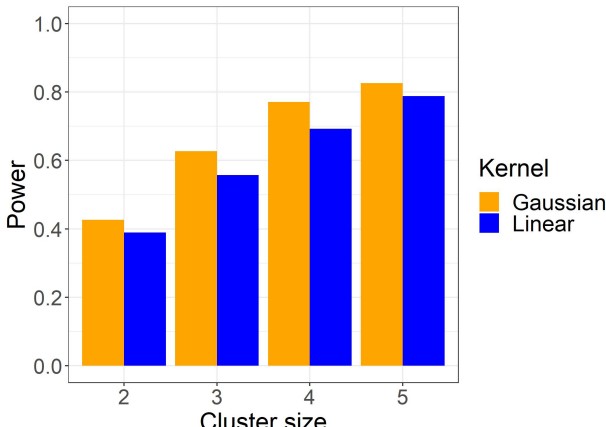

Figure F4: Empirical power of $\textbf{HSIC}_{\textbf{cl}}$ at nominal level $\alpha = 0.05$ under different cluster sizes. Simulation parameters are set as: $m = 500$, $\rho_c = 0.5$ and $\eta = 20\%$. **Power Scenario 1** is considered.

distribution. The p-value is calculated as

$$p_{\text{perm}} = \frac{\sum_{j=1}^{n_{\text{perm}}} I\{\text{HSIC}_{\text{perm}\,j} \geq \text{HSIC}_{\text{obs}}\}}{n_{\text{perm}}}, \tag{17}$$

where $n_{\text{perm}}$ is the number of permutations, $\text{HSIC}_{\text{perm}\,j}$ is the HSIC statistic at the $j$th permutation and $\text{HSIC}_{\text{obs}}$ is the original observed HSIC statistic.

We use the same simulation setting as in Section 4.1.1, where we consider a small sample size $m = 100$ and set $d = 3$ and $\rho_c = 0.5$. We compare $\textbf{HSIC}_{\textbf{cl}}$ against $\textbf{HSIC}_{\textbf{perm}}$ in both type I error control and power. In the power simulation, we consider **Power Scenario 1** and set $\eta = 20\%$. In each simulation run, 1000 permutations are conducted in $\textbf{HSIC}_{\textbf{perm}}$.

Figure F5 shows the p-value QQ-plots for $\textbf{HSIC}_{\textbf{cl}}$ and $\textbf{HSIC}_{\textbf{perm}}$ under Type I error simulation for both the Gaussian kernel and the linear kernel. Table F1 shows the empirical power of $\textbf{HSIC}_{\textbf{cl}}$ and $\textbf{HSIC}_{\textbf{perm}}$ for the two kernels.

Based on the Gaussian kernel, while $\textbf{HSIC}_{\textbf{cl}}$ is over-conservative, $\textbf{HSIC}_{\textbf{perm}}$ produces a valid type I error rate; as a result, $\textbf{HSIC}_{\textbf{perm}}$ has a higher power than $\textbf{HSIC}_{\textbf{cl}}$ in this case. Based on the linear kernel, $\textbf{HSIC}_{\textbf{cl}}$ and $\textbf{HSIC}_{\textbf{perm}}$ have similar performances. These results show that $\textbf{HSIC}_{\textbf{perm}}$ could be a useful surrogate for $\textbf{HSIC}_{\textbf{cl}}$ at small sample sizes. However, the computational burden of $\textbf{HSIC}_{\textbf{perm}}$ is large compared to $\textbf{HSIC}_{\textbf{cl}}$, especially as sample sizes increase (Table G1) or as we require a more stringent significance level (which requires a larger number of permutations).

Table F1: Empirical power of $\textbf{HSIC}_{\textbf{cl}}$ and $\textbf{HSIC}_{\textbf{perm}}$ at nominal level $\alpha = 0.05$ under simulation ($m = 100$).

| Kernel | $\textbf{HSIC}_{\textbf{cl}}$ | $\textbf{HSIC}_{\textbf{perm}}$ |
|---|---|---|
| Gaussian | 0.304 | 0.726 |
| Linear | 0.534 | 0.559 |

# G Additional details on implementation

## G.1 Code availability

In our simulations, the $\textbf{HSIC}_{\textbf{orig}}$ method is implemented according to Broadaway et al. [14] (called GAMuT in their work), where Davies' exact method [15] is used to approximate the mixture of chi-square variables in the asymptotic null distribution of HSIC. The specific code is adapted from `https://github.com/epstein-software/GAMuT` (license: GPL-3.0).

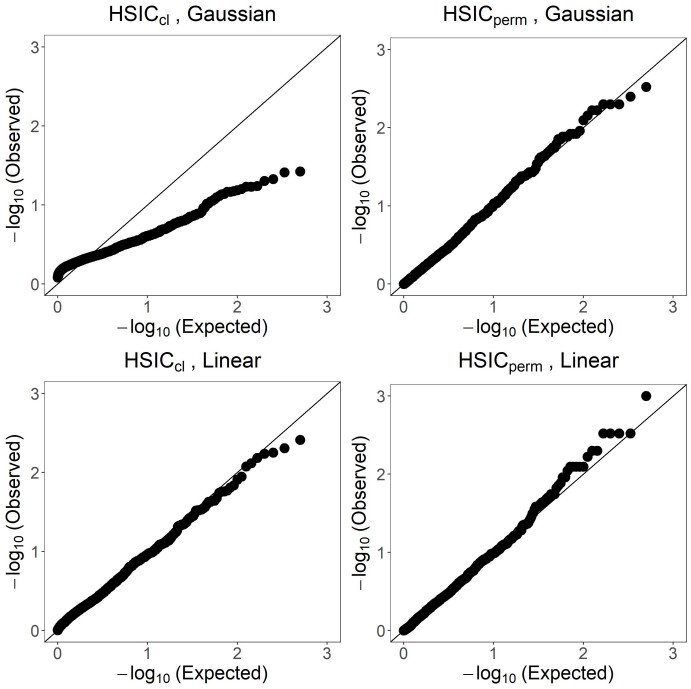

Figure F5: P-value QQ-plots for **HSIC$_{cl}$** and **HSIC$_{perm}$** under Type I error simulation. Simulation parameters are set as: $m = 100$, $d = 3$ and $\rho_c = 0.5$. The top row shows results based on the Gaussian kernel, and the bottom row shows results based on the linear kernel.

Both **HSIC$_{orig}$** and **HSIC$_{cl}$** use the CompQuadForm R package [16] v1.4.3: `https://cran.r-project.org/web/packages/CompQuadForm` (license: GPL $\geq$ 2), which implements Davies' exact method.

**HSIC$_{cl}$** is implemented as the `HSIC_{cl}()` function in R environment. In the Github page: `https://github.com/pearl-liu/HSIC_cl`, we provide code and instructions for using the `HSIC_{cl}()` function and for reproducing the simulation results in Section 4.1.2 of the main text.

### G.2 Computation time

We have estimated the computation time of **HSIC$_{cl}$** and **HSIC$_{perm}$** (with 1000 permutations; see Appendix F.3 for details) for different number of clusters ($m$). For each $m$, we simulate 10 data sets according to Section 4.1.1 of the main text and report the average computation time. Given constructed kernel matrices for $X$ and $Y$, the average computation times on a 12-core computer with 2.40 GHz CPUs and 256 GB memory are shown in Table G1.

Table G1: Average computation time (in seconds) of **HSIC$_{cl}$** and **HSIC$_{perm}$** (with 1000 permutations) for different number of clusters, with cluster size 3.

| Method | Kernel | Number of clusters, $m$ | | | | | |
|---|---|---|---|---|---|---|---|
| | | 50 | 100 | 200 | 500 | 800 | 1000 |
| **HSIC$_{cl}$** | Gaussian | 0.168 | 0.531 | 3.857 | 83.689 | 485.141 | 1109.086 |
| | Linear | 0.013 | 0.040 | 0.241 | 2.572 | 10.654 | 21.069 |
| **HSIC$_{perm}$** | Gaussian | 1.327 | 7.628 | 50.740 | 705.860 | 2802.158 | 5385.214 |
| | Linear | 1.392 | 7.158 | 49.335 | 692.443 | 2782.553 | 5353.857 |

First, we note that **HSIC$_{cl}$** has a much shorter computation time than **HSIC$_{perm}$** for each sample size, either based on the Gaussian kernel or based on the linear kernel.

Next we compare computation times of **HSIC$_{cl}$** between the Gaussian kernel and the linear kernel. For any $X \in \mathbb{R}^p$ with $p < n$, the associated linear kernel matrix has $p$ non-zero eigenvalues. In contrast, the Gaussian kernel matrix always has $n$ non-zero eigenvalues, due to the infinite dimension of its feature space. Therefore, based on the way the asymptotic null distribution of the HSIC statistic is estimated in Proposition 3.4, using a linear kernel can take much shorter computation times than using a Gaussian kernel. Specifically, the computational complexity of **HSIC$_{cl}$** based on a Gaussian kernel is $O(m^2 n^2)$. While Gaussian kernels have the advantage of capturing more general dependence patterns, linear kernels could be preferable in certain situations (e.g., when $p < n$) as a computationally efficient choice.

On a high-performance computing cluster (each node with 20 cores, 2.20 GHz CPUs and $\sim$100 GB memory), with divided computing jobs, it took $\sim$70 hours to complete the simulations in Section 4.1 using the Gaussian kernel, and $\sim$4 hours to complete the simulations using the linear kernel. Using the same resources, the analysis of the MsFLASH data set in Section 4.2 took $\sim$13 minutes.

# H MsFLASH study

## H.1 Description of the MsFLASH study

The Menopause Strategies: Finding Lasting Answers for Symptoms and Health (MsFLASH) Vaginal Health Trial was a randomized, double-blind and placebo-controlled clinical trial conducted at 2 centers in the U.S.: Kaiser Permanente Washington Health Research Institute in Seattle and University of Minnesota in Minneapolis [17]. The trial compared the treatment efficacy for moderate-to-severe vulvovaginal discomfort between 0.01 mg vaginal estradiol tablets or vaginal moisturizer and placebo in 302 postmenopausal women. Vaginal swabs were collected from the participants at baseline, and 4 and 12 weeks after randomization. In a secondary study [18], based on the samples collected from each follow-up, the vaginal microbiota was characterized via 16S ribosomal RNA (rRNA) gene sequencing, and the vaginal metabolome was profiled using liquid chromatography-mass spectrometry. The abundance data of microbial taxa were center log-ratio transformed to address differential read depth and compositionality, and the abundance data of metabolites were quantile-normalized. More details on the MsFLASH trial are described by Mitchell et al. (2018, 2021) [17, 18].

The MsFLASH trial was approved by institutional review boards of the participating institutions, and all participants provided written informed consent. Inspection of the data set reveals no personally identifiable information or offensive content. The data used in this study is available upon request from the MsFLASH Data Coordinating Center.

## H.2 Additional analysis results

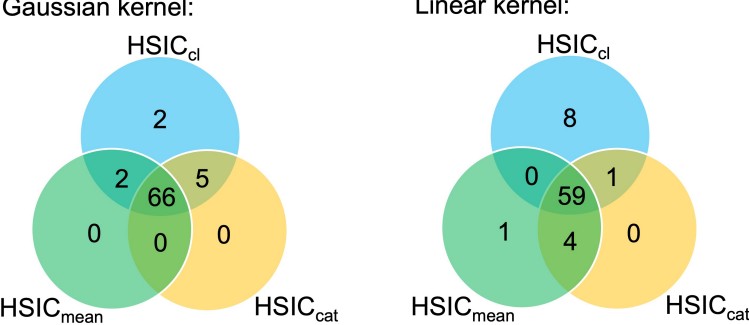

Figure H1: Venn diagrams for the number of metabolic pathways identified to be associated with the vaginal microbiome composition based on the MsFLASH data set ($\alpha = 5.3 \times 10^{-4}$). The results are separated by kernel.

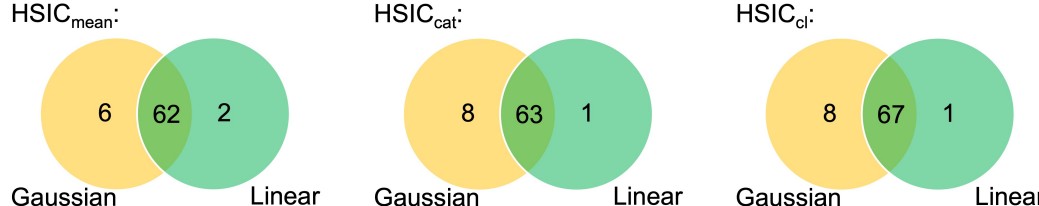

Figure H2: Venn diagrams for the number of metabolic pathways identified to be associated with the vaginal microbiome composition based on the MsFLASH data set ($\alpha = 5.3 \times 10^{-4}$). The results are separated by method for HSIC test.