# OpenReview forum: "A Kernel-based Test of Independence for Cluster-correlated Data"
_NeurIPS.cc/2021/Conference — NeurIPS 2021 Poster_

### Official Review · Reviewer_8fxs · 2021-07-05

**Rating:** 6
**Confidence:** 2

**Summary:**

The authors propose a test of independence for cluster correlated data. The test statistic is still HSIC. However they derive a different asymptotic distribution under cluster correlation.

**Limitations And Societal Impact:**

-

**Main Review:**

The paper seems well wrtitten. I don't have many comments about the writing  or the structure.

Unfortunately I believe the problem/contribution to be rather narrow. I am not sure if cluster correlated independence tests are impactful/ how an important this problem is.

In the real data experiment, could you write explictly what is X, what is Y, and what are their dimensions.

I DID NOT CHECK PROOFS IN THE APPENDIX. I think 14 pages of technical appendix is challenging to completely verify. I believe it would be stronger if most of the proof could be given in the main text, or some insightful high-level discussion/sketch could be given that suggests the proof is correct.

Experimental evidence that the results are correct are currently not very strong I believe. For the Gaussian kernel the p-values are not that close to alpha (although they are smaller so in terms of type 1 error it seems ok). Could you experimentally demonstrate the p-values are uniformly distributed under the null? Or take larger samples?

While it is nice there is a check fo the type 1 error experimentally in table 1, I'm not sure how convincing one gaussian distribution is to conclude from this that the type 1 error rate is correct.

Overall view:

The paper is generally well written and structured. The experiments are well organized and described. I'm a bit in doubt between weak accept/weak reject. The proofs are all in the appendix and I was not able to check them due to length. More insightful proof sketch that suggests the analysis is correct/ more extensive type 1 experiments could help in that respect. Main question for me is if cluster correlated independence tests are relevant/impactful enough for neurips.



**Time Spent Reviewing:**

5

---

> ### Author Response · Authors · 2021-08-10
> **Author response to Reviewer 8fxs**
>
> Thank you for the suggestions and comments. We will provide more discussion and intuition on the proof in the main text. Here are our detailed responses in regard to your concerns.
>
> 1. Relevance of cluster-correlated independence tests
>
> We believe that the problem we are tackling is of broad relevance. Dependence measures and independence tests have been utilized in many statistical and machine learning areas, such as constructing causal networks and performing feature selection (where the dependence between features and labels is maximized). HSIC and related approaches are now finding their way into the applied literature such as biomedical studies, where they have culminated in many new discoveries. On the other hand, clustered data are very common in different scientific fields. The best example of clustered data is longitudinally collected data, which is ubiquitous in epidemiological studies, social sciences, consumer research, etc. Other types of clustered correlation can arise in areas such as educational studies (classroom and school effects) and environmental studies (spatial correlation). Our work represents an initial foray into enabling a direct application of the HSIC to correlated data from these numerous fields of study. Therefore, we expect that an HSIC-based test for correlated data would be an impactful contribution with a wide range of practical applications.
>
> 2. Real data experiment
>
> Here X represents the abundance of microbial taxa, with dimension 381. Y represents the abundance of metabolites within a specific metabolic pathway, with dimension corresponding to the number of metabolites in that pathway (ranging from 1 to 45 in this data set).
> We will make this clearer in the final paper.
>
>
> 3. Control of type I error rate
>
> Based on p-value QQ-plots, we have observed a closer alignment of the p-values to the uniform distribution as the number of clusters ($m$) gets larger, when using the Gaussian kernel in type I error simulation. We will include additional simulation results with larger $m$. As a preliminary result, we have observed that, at $m$ = 1500 and $\rho_c$ = 0.7, the type I error rate is close to the nominal alpha for the Gaussian kernel.
>
> In the simulation study, we generated data using the multivariate Gaussian distribution as it is convenient to incorporate within-cluster correlation with this distribution. We will continue to explore other simulation settings with different distributions and include additional type I error experiments.

---

### Official Review · Reviewer_WwvK · 2021-07-09

**Rating:** 7
**Confidence:** 3

**Summary:**

This submission introduces a calibrated testing procedure for the
independence between two variables under non-iid sampling. The
sampling is assumed to follow a clustered structure, where samples
within clusters are correlated but the clusters are independent from
each others. The procedure relies on the Hilbert-Schmidt Independence
Criterion (HSIC), and a first contribution is an asymptotic null
distribution under the clustered sampling assumption. The test is also
shown to be consistent. Experiments on simulated and real data show
that the proposed test guarantees a control on the type I error, and
has higher power than existing alternatives in a variety of scenarios.


**Limitations And Societal Impact:**

The authors adequately addressed the limitations and potential negative societal impact of their work.

**Main Review:**

The submission is clear and well written. The problem of non-iid data
is an important and (in my opinion) under-addressed problem in
statistics and machine learning, making this work a useful and
interesting contribution. This contribution is significant as the
proposed results are non-trivial. The experiments are thorough and
illustrate the proposed results well.

I would appreciate some clarification on the effect of d (the size of
the clusters) on the proposed results. The sampling assumption
constrains all clusters to be of equal size d, so that the sample size
n is m*d where m is the number of clusters. All asymptotics are given
in m->inf (and therefore n->inf) and these asymptotics don't seem to
depend on d. Is it possible to say something about non-asymptotic
properties of the statistic that would vary with the cluster size?
Does this size affect the convergence speed? Experiments with varying
cluster sizes would also be welcome. With d=1 one recovers the iid
setting, and all experiments are done with d=3. Larger d will always
increase the dependence. Would the performance (calibration, power) of
the proposed procedure be affected in this case?

I also have a few minor comments:

- I think that the statement of Proposition 3.4 could be made more
precise. It currently states that under some assumptions, "we can
compare the statistics [...] against the (1-alpha) quatile of the
distribution [...]". I would find it more clear if the convergence
result was stated in the Proposition (and the consequence on practical
testing could be discussed after the Proposition).

- line 129, the "centered kernel functions" could be defined more
  clearly. The only centering defined so far has been on empirical nxn
  Gram matrices, not on functions.

- lines 234-236, I couldn't understand how X** differed from X. Both
  are defined as (x_11, x_12, x_13, ..., x_p1, x_p2n x_p3). If they
  are identical, this should be stated clearly. And in this case,
  HSIC_cat is equivalent to HSIC_orig?

- line 254, it could be useful to provide intuition as to why HSIC_cl
  with Gaussian kernel tends to be conservative.

- line 254, it is not clear from the table that the type I error rate
  gets closer to the nominal alpha when m=1000, the difference with
  m=500 is tiny. Did you observe a stronger improvement with larger m?

- Figure 1: power is provided at nominal level for all methods. Are
  the baseline methods correctly calibrated?


**Time Spent Reviewing:**

4

---

> ### Author Response · Authors · 2021-08-10
> **Author response to Reviewer WwvK**
>
> Thank you for the suggestions and comments. Here are our detailed responses.
>
> 1. Effect of cluster size on HSIC_cl
>
> Under our assumption of fixed cluster size $d$, the asymptotic results we developed here hold regardless of the value of $d$. However, as you suggested, we believe it would be interesting to investigate the effect of cluster size on convergence speed and test performance. We have conducted some preliminary simulations with varied cluster sizes. From what we have observed, when the number of clusters ($m$) and the level of within-cluster correlation ($\rho_c$) are fixed, a larger cluster size tends to result in a higher statistical power, possibly due to increase in the overall sample size and information gain. Type I error control tends to be similar for different cluster sizes, suggesting that the convergence speed is likely not affected by cluster size.
>
> Interestingly, the performance of HSIC_cl in statistical power not only depends on the cluster size $d$, but also depends on the within-cluster correlation $\rho_c$: For smaller $\rho_c$, we will have a larger effective sample size, resulting in a greater power gain. This is also shown in our power simulation results (Figure 1 and 2).
>
> We will include additional simulation results and discussion regarding different cluster sizes in our final paper.
>
> 2. Centered kernel function
>
> A centered kernel function $\tilde{k}$ for kernel $k$ is defined as:
>
> $$\tilde{k}(x, x^\prime)= k (x, x^\prime) - E_X [k (X, x^\prime)] - E_{X^\prime} [k (x, X^\prime)] + E_{XX^\prime}[k (X, X^\prime)]. $$
>
> We will explicitly define this in our final paper.
>
>
> 3. Difference between X** and X in simulation
>
> X** = (x_11, x_12, x_13, ..., x_p1, x_p2, x_p3) is defined at the cluster level and has dimension $3p$. It contains all 3 observations within a cluster.
>
> X is defined at the observation level and has dimension $p$. For example, the vector (x_11, ..., x_p1) would be one observation of X, and (x_12, …, x_p2) would be another observation of X.
>
> HSIC_cat is performed by applying the original i.i.d.-data-based HSIC test to the $m \times 3p$ data matrix composed of cluster-level data X**'s, while HSIC_orig is performed by applying the original HSIC test to the $3m \times p$ data matrix composed of observations of X's.
>
>
> 4. Conservativeness of Gaussian kernel
>
> In the proofs for the asymptotics of HSIC (both the original HSIC and HSIC_cl), the test statistic converges asymptotically to a weighted sum of independent squared standard-normal variables (Chi-square variables) under the null. However, in finite samples with small sample sizes, it has been shown that [1] this kind of test statistic is instead close in distribution to a weighted sum of negatively correlated squared normal variables that have variance smaller than 1. Since the Gaussian kernel is associated with $n$ non-zero eigenvalues (whereas the linear kernel has fewer) in finite samples, the null distribution based on the Gaussian kernel includes more terms in the weighted sum (as the weights depend on  eigenvalues), which could in turn aggregate more errors, compared to the linear kernel. Hence the test statistic based on the Gaussian kernel has a slower convergence to the asymptotic distribution. We will continue to investigate this issue and better describe this in the paper.
>
>
> 5. Control of type I error rate for Gaussian kernel
>
> Based on p-value QQ-plots, we have observed a closer alignment of the p-values to the uniform distribution as the number of clusters gets larger, when using the Gaussian kernel in type I error simulation. We will include additional simulation results with larger $m$. As a preliminary result, we have observed that, at $m$ = 1500 and $\rho_c$ = 0.7, the type I error rate is close to the nominal alpha for the Gaussian kernel.
>
>
> 6. Calibration of baseline methods
>
> The two competing/baseline methods are performed by applying the original i.i.d.-data-based HSIC test to the cluster-level data. The original HSIC test is correctly calibrated in terms of having valid type I error rates for i.i.d data at the sample sizes we considered.
>
> References:
>
> [1] Seunggeun Lee, Mary J Emond, Michael J Bamshad, Kathleen C Barnes, Mark J Rieder, Deborah A Nickerson, ESP Lung Project Team, David C Christiani, Mark M Wurfel, Xihong Lin, et al. Optimal unified approach for rare-variant association testing with application to small-sample case-control whole-exome sequencing studies. The American Journal of Human Genetics, 91(2):224–237, 2012.

---

> > ### Comment · Reviewer_WwvK · 2021-08-26
> > **Response to the rebuttal**
> >
> > Dear authors,
> >
> > Thank you for taking the time to write this thorough and detailed response. It answered all of my questions.
> >
> > All of the best.

---

### Official Review · Reviewer_wjAG · 2021-07-16

**Rating:** 7
**Confidence:** 3

**Summary:**

The paper derives an approximation of the asymptotic null distribution of the Hilbert-Schmidt Independence Criterion under a matched and complete clustering. The null distribution corresponds to independence between the variables.  The samples consist of equally sized clusters each with matched within cluster correlation. The distribution is described as a weighted sum of chi-squared variables, where the weights are approximated by the eigenvalues of a matrix composed from the eigenstructure of the marginal kernel matrices grouped by cluster. Simulations confirm the increased power of the method. Specifically, simulation shows the test is conservative in lower sample size, but increases in power versus other methods (concatenating the samples longitudinally).  Experimental results demonstrate the increased power is able to identify additional metabolite pathways are associated with the vaginal microbiomes composition.

**Limitations And Societal Impact:**

A statement of the computational complexity and its comparison to the computation required to run permutation tests with HSIC would be useful.

**Main Review:**

**Originality** This is a useful tool for dependence testing in cases of non-i.i.d. instances. The paper's contribution is useful. The paper builds from Zhang et al. taking the assumption of equivalent correlation within each cluster structure to derive the novel asymptotic distribution.

**Quality** The technical description and experiment are statistically sound. The discussion regarding small sample methods (non-asymptotic) could be expanded beyond the reference to Zhan et al. (2017).

In terms of comparisons, it seems permutation based tests could derive surrogate null distributions to handle the cases of missing data and unequal cluster size.  Especially for the simulated data, I would like to see a comparison versus permutation based methods. On the real data, it seems that the  stringent significance threshold (based on Bonferroni correction) would require a large number of permutations, but it could be useful to compare.

**Clarity** The paper is well-organized and well-written.

**Significance** A solid statistical test for dependence has the potential to impact scientific discovery in exploratory work such as biomarker discovery. Longitudinally, clustered data is increasing in importance in biomedical health applications.  A properly documented and released code may positively impact biostatistical practice.

### After authors' response.
I have read the authors' response and the other reviews. I maintain my overall score. I hope the authors can report computational complexity (cubic in the sample size to determine asymptotic distribution for the Gaussian kernel) could be mentioned in addition to the wall clock time.


**Time Spent Reviewing:**

3.5

---

> ### Author Response · Authors · 2021-08-10
> **Author response to Reviewer wjAG**
>
> Thank you for the suggestions and comments. Here are our detailed responses.
>
> 1. Comparison against permutation-based methods
>
> We agree that comparisons with permutation-based methods would be useful, and indeed, we have previously conducted some preliminary simulations. From what we observed, with a sufficient number of permutations, the permutation-based HSIC is able to yield valid type I error rates and achieve a greater power than HSIC_cl at small sample sizes -- as HSIC_cl is conservative when sample sizes are small. However, the computational burden of permutation-based HSIC would be very large as sample sizes increase or as we require a more stringent significance level. For example, in an experiment with 1000 pathways, the alpha level would be 0.05/1000 = $5 \times 10^{-5}$, requiring at least $10^6$ permutations to get an accurate p-value, where each permutation would require working with $n \times n$ kernel matrices. In short, we expect permutation-based methods to be good surrogates for asymptotics-based methods in the case of small sample sizes, though they would be difficult to apply in situations where sample sizes are larger and where one is interested in accurately estimating small p-values.
>
> 2. Information on computation time
>
> This is an excellent point that will help illustrate the importance of our work. The computation time for HSIC_cl has been provided in Appendix F.2. We will include additional information on computation time for permutation tests.

---

### Official Review · Reviewer_myr6 · 2021-07-16

**Rating:** 6
**Confidence:** 3

**Summary:**

The paper proposed a modified version of Hilbert-Schmidt Independence Criterion (HSIC) to test the dependence between two multivariate variables. The method is suitable for data that is not iid, i.e. correlation exists between different data points. The paper first demonstrate the asymptotic distributions of HSIC under the null hypothesis (independent) and the condition for the alternative in Theorem 3.2 and 3.3. The approximation of the asymptotic HSIC distribution is provided in proposition 3.4. In the simulation, the proposed method HSICcl shows a better performance than earlier versions of HSICorig both in terms of Type I error and empirical power. In real data, HSICcl identifies more associated pathways than previous versions.


**Limitations And Societal Impact:**

yes

**Main Review:**

I have not be able to go through the theoretical part and just comment on the real experiment part. It will be nice to justify the identified pathways are biologically meaningful, especially the newly identified pathways. In line 284, it says “The original study found no additional benefit of estradiol or moisturiser over placebo.” How is this conclusion linked with results provided in this paper? Also, it says “In a second study …..”, what is the main conclusion of that paper? Is the main conclusion supported by this paper’s results?

I think this test will be interesting if its predictions could be justified in the biological setting.

---------------------------
I am convinced by the authors' reply that the extra detected pathways are biologically sensible and I will raise my rating for this paper.

**Time Spent Reviewing:**

7

---

> ### Author Response · Authors · 2021-08-10
> **Author response to Reviewer myr6**
>
> Thank you for the suggestions on the real data analysis. Here are our detailed responses.
>
> 1. Biological significance of identified pathways
>
> We agree that further justification of the biological findings would be helpful and will include more discussion on biological implications of our findings in the final paper.
>
> As an example, here we focus on some of the top pathways (with high statistical significance) identified using our proposed method and highlight their biological relevance. The top pathways include multiple metabolism pathways for amino acids. In healthy women, the vaginal microbiome is dominated by the Lactobacillus genus, which maintains the vaginal homeostasis and prevents the growth of adverse microorganisms. Lactobacillus species are known to produce branched-chain amino acids such as valine, leucine, and isoleucine [1]. All these amino acids are present in our top pathways. In particular, one pathway related to leucine metabolism is only identified by our proposed method but not by the other two competing methods. Therefore, our finding is consistent with previous studies on bacterial metabolism. It could also form the basis for future investigation of these amino acids as potential biomarkers for vaginal health status.
>
> 2. Relation of our analysis to previous studies on the MsFLASH trial
>
> The original study (Study 1) on the MsFLASH trial aimed to evaluate the treatment effect of vaginal estradiol (estrogen) vs. placebo on vaginal discomfort in postmenopausal women. A central hypothesis of the trial was that introducing estradiol would encourage beneficial shifts in the vaginal microbiome, which would in turn lead to reduced discomfort. Unfortunately, despite considerable preliminary evidence and years of efforts, the original trial failed to find a beneficial treatment effect of estradiol. Therefore, ancillary studies were conducted to understand why the intervention failed and provide clues as to potential novel treatments.
>
> Accordingly, a second study (Study 2) based on the trial data was conducted to evaluate whether estrogen indeed shifted the microbiome composition. The main results in Study 2 show that the vaginal microbiome was shifted by the estrogen therapy, but this shift did not result in improved symptoms for the women in the trial. Therefore, a further effort is to understand whether there is a breakdown in the link between the microbes and metabolites (thought to be produced by microbes). A breakdown in this link would indicate a disruption in normal metabolic functions of the microbes, which might explain the lack of association between shift in microbiome composition and symptom improvement. Our work fills a critical gap by facilitating this exact analysis: we have observed strong correlations between the vaginal microbiome and metabolites. Thus, the failure of the estrogen treatment is likely not due to disruption in the microbe-metabolite link and lies elsewhere. Further studies would be needed to uncover the biological mechanisms of postmenopausal vaginal discomfort.
>
> We will include a more detailed context for the MsFLASH trial in our final paper.
>
> References:
>
> [1] Camilla Ceccarani, Claudio Foschi, Carola Parolin, Antonietta D’Antuono, Valeria Gaspari, Clarissa Consolandi, Luca Laghi, Tania Camboni, Beatrice Vitali, Marco Severgnini, et al. Diversity of vaginal microbiome and metabolome during genital infections. Scientific reports, 9(1):1–12, 2019.

---

### Decision · Program_Chairs · 2021-09-27

**Decision:**

Accept (Poster)

**Comment:**

The paper proposes a test of independence for cluster correlated data based on HSIC and derive an asymptotic distribution for HSIC under cluster correlation.
In evaluations, the test is slightly conservative.
The test is applied to a microbiome study.
The paper is well organized and well written.
After discussion with the authors, only minor issues remain.

However, the reviewers agree that this is a good contribution and should be accepted.